# COST-OF-PASS: AN ECONOMIC FRAMEWORK FOR EVALUATING LANGUAGE MODELS

**Mehmet Hamza Erol**[*]
Stanford University
mhamza@stanford.edu

**Batu El**[*]
Stanford University
batu@stanford.edu

**Mirac Suzgun**[*]
Stanford University
msuzgun@stanford.edu

**Mert Yüksekgönül**[†]
Stanford University
mertyg@stanford.edu

**James Zou**[†]
Stanford University
jamesz@stanford.edu

## ABSTRACT

The widespread adoption of AI systems in the economy hinges on their ability to generate economic value that outweighs their inference costs. Evaluating this tradeoff requires metrics that account for both performance and costs. Building on Farrell's theory of productive efficiency, we develop an economically grounded framework for evaluating language models' productivity by combining accuracy and inference cost. We formalize *cost-of-pass*, the expected monetary cost of generating a correct solution. We then define the *frontier cost-of-pass* as the minimum cost-of-pass achievable across available models or the human-expert(s), using the approximate cost of hiring an expert. Our analysis reveals distinct economic insights. *First*, lightweight models are most cost-effective for basic quantitative tasks, large models for knowledge-intensive ones, and reasoning models for complex quantitative problems, despite higher per-token costs. *Second*, tracking this frontier cost-of-pass over the past year reveals significant progress, particularly for complex quantitative tasks where the cost has roughly halved every few months. *Third*, to trace key innovations driving this progress, we examine counterfactual frontiers—estimates of cost-efficiency without specific model classes. We find that innovations in lightweight, large, and reasoning models have been essential for pushing the frontier in basic quantitative, knowledge-intensive, and complex quantitative tasks, respectively. *Finally*, we assess the cost-reductions from common inference-time techniques (majority voting and self-refinement), and a budget-aware technique (TALE-EP). We find that performance-oriented methods with marginal performance gains rarely justify the costs, while TALE-EP shows some promise. Overall, our findings underscore that complementary model-level innovations are the primary drivers of cost-efficiency, and our economic framework provides a principled tool for measuring this progress and guiding deployment. [*]

## 1 INTRODUCTION

The recent progress in generative AI, particularly language models (LMs), has sparked significant interest in their potential to transform industries, automate cognitive tasks, and reshape economic productivity (Brynjolfsson et al., 2025; Eloundou et al., 2024; Acemoglu, 2024). The widespread adoption of these AI systems in the economy hinges on whether the economic benefits generated by the tasks they can perform outweigh the associated inference costs, and whether those inference costs are lower than the cost of equivalent human labor. Consequently, two priorities have emerged at the forefront of LM research: advancing capabilities and reducing costs. These goals, however, often involve trade-offs with more powerful models or test-time techniques that offer higher accuracy at the expense of greater computational and monetary cost (Chen et al., 2024; Parashar et al., 2025; Madaan et al., 2023; Wang et al., 2023; Kapoor et al., 2024). While standard metrics capture accuracy or other system capabilities, they fail to account for cost, leading to an incomplete picture of progress.

---

[*]Co-first authors.    [†]Co-senior authors.    [*] https://github.com/mhamzaerol/Cost-of-Pass.

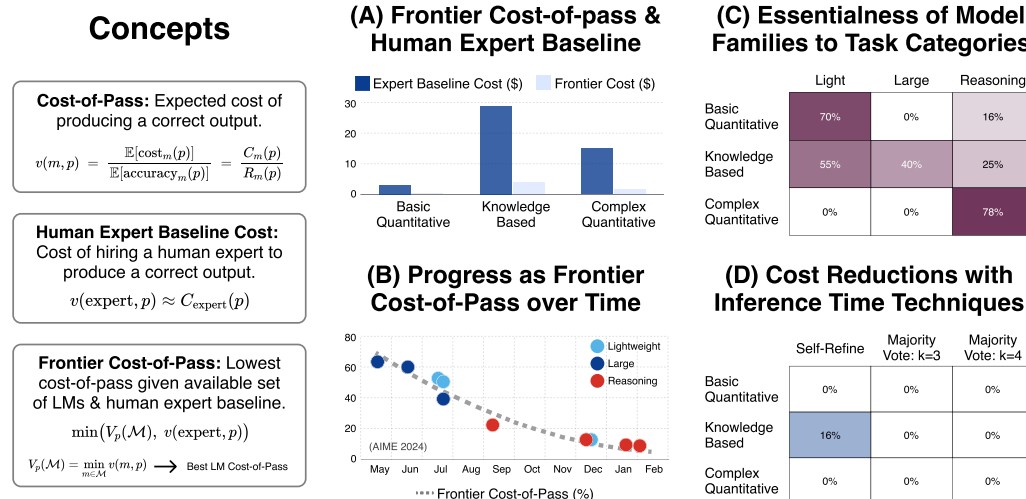

Figure 1: **Highlights of the cost-of-pass framework and empirical analyses.** Core concepts (left) set foundations for: **(A)** Comparing the Human Expert Baseline to the frontier achieved by the single most effective LM per task category. **(B)** Tracking the reduction in frontier cost-of-pass over time, indicating progress driven by new model releases (color-coded by family). **(C)** Quantifying the essential contribution of each model family: lightweight (less than $1 per million tokens), large, and reasoning; to the current cost-efficiency frontier, measured by the percentage of each family's contribution. **(D)** Assessing the economic benefit (relative cost reduction) achieved by applying common inference-time techniques over the baseline model frontier (which rarely results in meaningful gains).

Ultimately, what matters to the users is not just raw capability, but the value delivered relative to cost and the standard has been to interpret and report these separately. As the ecosystem of models grows, it is essential to assess new models not in isolation, but in the context of a broader ecosystem, where marginal improvements may or may not justify higher costs, and do so in an easy-to-interpret manner.

Prior attempts to incorporate cost into evaluation have typically relied on fixed inference budgets (Wang et al., 2024), heuristic performance scores (McDonald et al., 2024), or non-monetary proxies such as conciseness (Nayab et al., 2024). These choices tie conclusions to specific constraints or heuristics, limiting the generality and economic interpretability of such evaluations. To systematically investigate the trade-off between cost and performance and analyze the LM ecosystem as a whole, we draw insights from a well-established and foundational framework from economics: production frontiers. Economists have long studied these frontiers, which map a set of inputs to *the maximum output attainable under a given technology* (Farrell, 1957). In Farrell's original formulation, a producer is *technically efficient* if no input can be reduced without lowering output, and *price efficient* if the input mix minimizes cost given input prices. Together, these conditions yield the lowest possible cost per unit of output. Extending this framework, Aigner et al. (1977) introduced stochastic frontier production functions, in which the relationship between inputs and output is modeled as stochastic rather than deterministic, practically accounting for potential defective outputs that do not pass evaluation criteria due to factors beyond the producer's control.

These economic concepts are highly relevant to modern LMs, which inherently function as stochastic producers: for a given input, they yield a desired output (e.g., a correct solution) stochastically (Brown et al., 2024). Common practices such as employing scaffolds or more computationally intensive inference techniques (Snell et al., 2024; Madaan et al., 2023; Wang et al., 2023) represent efforts to manipulate this production process. These strategies seek to increase the probability of success but typically do so at the expense of higher computational cost, directly mirroring the economic trade-offs inherent in production efficiency. Motivated by these parallels and the economic goal of minimizing cost per successful output under uncertainty, we adapt the ideas from economically grounded production theory to develop a quantitative evaluation framework tailored to LMs.

We summarize our contributions as follows.

**Concepts.** We introduce *cost-of-pass* (§2.2), which quantifies the expected monetary cost to achieve a successful output for a given problem. Building on this concept and incorporating a human-expert

cost baseline, we define the *frontier cost-of-pass* (§2.4) as the minimum achievable cost-of-pass across all available options (LMs and human-expert) for that problem. We show these reveal distinct economic niches for model families (e.g., lightweight *vs.* reasoning models) on different tasks, which accuracy comparisons alone obscure (§3.2).

**Tracking progress with frontier cost-of-pass.** Using the cost-of-pass and frontier cost-of-pass, we analyze economic improvements across three task categories from May 2024 to February 2025. We observe an exponential decrease in frontier cost-of-pass across all tasks, though the trends vary. Notably, we observe that, over the past year, the expected cost of generating a correct solution to complex quantitative problems has been cut in half every few months. We find that the frontier cost-of-pass is driven primarily by lightweight models and reasoning models (§3.3).

**Counterfactual frontier in the absence of model families.** We show that our analysis reveals the complementary roles of different model types in driving recent progress. Innovations in lightweight models have been instrumental in reducing costs on basic quantitative tasks. Large models, by contrast, have been most impactful for knowledge-based benchmarks like GPQA-Diamond (Rein et al., 2024). Meanwhile, reasoning models have been central to advances on complex quantitative reasoning challenges such as AIME (Mathematical Association of America, 2024) and MATH (Hendrycks et al., 2021) (§ 3.4).

**Impact of post-hoc inference time techniques.** We observe that common test-time techniques such as self-refinement (Madaan et al., 2023) and majority voting (self-consistency; Wang et al., 2023) to improve performance offer either limited or no economic benefits, while a budget-aware technique TALE-EP (Han et al., 2025) delivers some benefits. These indicate that the recent reductions in frontier cost-of-pass have been mostly driven by model-level innovations (§ 3.5).

## 2 SETUP

### 2.1 ECONOMIC THEORY OF PRODUCTION EFFICIENCY

Classical production theory examines how producers efficiently convert inputs (resources) into outputs. A central concern is understanding the maximum output attainable with a given set of inputs, or conversely, the minimum inputs (and thus cost) required to achieve a specific target output level.

Consider a set of producers $\mathcal{F} = \{f_i\}_{i=1}^n$ such that each producer $f_i \in \mathcal{F}$ can transform an input vector $\mathbf{x} \in \mathbb{R}_{\geq 0}^k$ (e.g., quantities of different resources) into an output. The inputs used by producer $f_i$ have associated prices, represented by a price vector $\mathbf{w_i} \in \mathbb{R}_{\geq 0}^k$. When focusing on achieving a specific target output level, say $u$ units, economists are interested in the *frontier cost* $V_u$. This represents the absolute minimum monetary cost required to produce at least $u$ units, considering all input vectors $\mathbf{x}$ capable of achieving this output across all available producers with their respective pricings. This frontier cost is formally defined as:

$$V_u = \min_{f_i \in \mathcal{F}} \left\{ \mathbf{w}_i^\top \mathbf{x} \,\middle|\, f_i(\mathbf{x}) \geq u \right\}, \tag{1}$$

Farrell (1957) used these core concepts to formalize definitions for technical and price efficiency in a production ecosystem for producers. Critically, Aigner et al. (1977) extended this framework to handle *stochastic* production functions, where output is probabilistic for a given input.

Building on this economic foundation, we adapt the core concept of a frontier cost ($V_u$) to represent the minimum achievable cost for obtaining a correct solution using LMs. To better reflect LM behavior, which is inherently stochastic, we incorporate this variability into our cost-efficiency metric. This aligns our framework with core production concepts and enables assessment of the economic impact of stochastic LM producers.

### 2.2 COST-OF-PASS: AN EFFICIENCY METRIC FOR LMS

Here we instantiate the economic framework for LMs. Consider a specific *problem* $p$, where the unit of production is a correct solution. We define a *model* $m$ as an inference pipeline using an LM, acting

as a stochastic producer. Two quantities characterize its efficiency on problem $p$:

$$R_m(p) = \text{Prob. of } m \text{ producing a correct answer on } p,$$
$$C_m(p) = \text{Expected cost of one inference attempt by } m \text{ on } p.$$

In the context of LMs, the inputs $\mathbf{x}$ correspond to resources like prompt and generated tokens, while the input prices $\mathbf{w}$ represent the costs per token charged by the provider. The total cost of these inputs for a single inference attempt by model $m$ on problem $p$ is captured by $C_m(p)$, effectively instantiating the term $\mathbf{w}^\top \mathbf{x}$ from the theory in the previous section.

Since the model output is stochastic, the expected number of attempts to obtain the first correct solution is $1/R_m(p)$, assuming independent trials. This yields the ***cost-of-pass***, defined as the expected monetary cost to obtain one correct solution for problem $p$:

$$v(m, p) \;=\; \frac{C_m(p)}{R_m(p)}. \tag{2}$$

The cost-of-pass integrates both performance ($R_m(p)$) and cost ($C_m(p)$) into a single economically interpretable metric: it quantifies how efficiently financial resources are converted into correct outputs. This formulation mirrors classical production theory, where the goal is to assess the cost of achieving a specific target output (Farrell, 1957); in our case, the target is a correct solution. When a model cannot produce one ($R_m(p) = 0$), the cost-of-pass becomes infinite, appropriately signaling infeasibility.

### 2.3 The LM Frontier Cost-of-Pass

While cost-of-pass (§ 2.2) evaluates a single model's efficiency, understanding the overall state of LM capabilities for a given problem requires assessing the collective performance of the entire available LM ecosystem. Therefore, analogous to the frontier cost $V_u$ (Eq. 1), we define the *LM frontier cost-of-pass* for problem $p$ as the minimum cost-of-pass achievable using any available LM strategy $m$ from the set $\mathcal{M}$:

$$V_p(\mathcal{M}) = \min_{m \in \mathcal{M}} v(m, p). \tag{3}$$

$V_p(\mathcal{M})$ quantifies the minimum expected cost to solve problem $p$ using the most cost-effective model currently available within the set $\mathcal{M}$. If no LM in $\mathcal{M}$ can solve $p$ (i.e., $R_m(p) = 0$ for all $m \in \mathcal{M}$), then $V_p(\mathcal{M}) = \infty$.

### 2.4 Grounding Evaluation: Estimated Human-Expert Baseline

The LM frontier cost-of-pass $V_p(\mathcal{M})$ reveals the best LM performance but lacks context: it does not show if LMs are economically advantageous over human labor. Moreover, the LM frontier cost-of-pass can be infinite if no LM succeeds. To address both, we introduce *human-expert baseline* as a reference point, by considering a human-expert annotator as a specific strategy: $m_{\text{expert}}$. Let $\mathcal{M}_0 = \{m_{\text{expert}}\}$ represent this baseline set. We assume experts typically achieve near-perfect correctness ($R_{\text{expert}}(p) \approx 1$) for tasks they are qualified for[†]. Thus, the cost-of-pass for a qualified expert is approximately their labor cost per problem:

$$v(\text{expert}, p) \approx C_{\text{expert}}(p). \tag{4}$$

The estimation of $C_{\text{expert}}(p)$ involves considering required expertise, time per problem, and appropriate compensation rates (detailed in § 2.6.1). By incorporating this baseline, we define the *frontier cost-of-pass* for problem $p$, considering both LMs ($\mathcal{M}$) and the human-expert alternative ($\mathcal{M}_0$):

$$V_p(\mathcal{M} \cup \mathcal{M}_0) = \min\big(V_p(\mathcal{M}), \; v(\text{expert}, p)\big). \tag{5}$$

This frontier cost-of-pass represents the true minimum expected cost to obtain a correct solution for problem $p$ using the best available option, whether it's an LM or a human. Crucially, $V_p(\mathcal{M} \cup \mathcal{M}_0)$ is always finite (assuming finite human-expert cost and capability).

---

[†] The qualified expert is intended to match the benchmark's label creators, for whom we implicitly assume near-perfect correctness. Appendix D.3 discusses associated limitations with possible relaxations.

## 2.5 Measuring Progress and Value Gain

To track improvements against the best available option over time, let $\mathcal{M}_t$ denote the *total set of available strategies* at time $t$, encompassing both the set of LM strategies released up to time $t$ *and* the human-expert baseline $\mathcal{M}_0$, that is, $\mathcal{M}_t = \{m_{\leq t}\} \cup \mathcal{M}_0$. The frontier cost-of-pass achievable at time $t$ can be calculated as:

$$V_p(\mathcal{M}_t) = \min_{m \in \mathcal{M}_t} v(m, p). \tag{6}$$

As new LM models $\{m_t\}$ are released, the set expands such that $\mathcal{M}_t = \mathcal{M}_{t-1} \cup \{m_t\}$. Consequently, the frontier cost-of-pass $V_p(\mathcal{M}_t)$ forms a *non-increasing* sequence over time $t$, tracking the reduction in the minimum cost needed to solve a particular problem $p$.

To quantify the economic impact of new developments, we define the *gain*. When a new set of models $\{m_t\}$ becomes available at time $t$ (often a single model), the gain for problem $p$ is the reduction it causes in the frontier cost-of-pass:

$$G_p(\{m_t\}, \mathcal{M}_{t-1}) = V_p(\mathcal{M}_{t-1}) - V_p(\mathcal{M}_{t-1} \cup \{m_t\}). \tag{7}$$

Note that $G_p$ measures how much cheaper the new model(s), $\{m_t\}$, make solving $p$ compared to prior best options, including humans. Hence, a large $G_p$ value indicates a significant economic contribution in solving $p$. This notion underlies our experiments, analyzing the value generated by models relative to the human baseline and tracking the evolution of the overall frontier.

**Extending to a distribution.** Although measuring frontier cost-of-pass and value gain for individual problems can be informative, particularly through a fine-grained perspective, we often care about more than a single instance. Let $P = \{p_i\}_{i=1}^n$ be $n$ problems drawn i.i.d. from $D$. We treat $P$ as the empirical distribution that puts mass $1/n$ on each element. We can then extend our definitions for such a distribution through the following:

$$V_{p \sim D}(\mathcal{M}_t) \approx \mathbb{E}_{p \sim P}[V_p(\mathcal{M}_t)], \tag{8}$$

$$G_{p \sim D}(\{m_t\}, \mathcal{M}_{t-1}) \approx \mathbb{E}_{p \sim P}[G_p(\{m_t\}, \mathcal{M}_{t-1})]. \tag{9}$$

## 2.6 Estimating the Economic Efficiency

To operationalize our overall framework for any given distribution of problems, we introduce the following recipe:

(1) **Estimate success rates.** For each model-problem pair $(m, p)$, generate a number of independent attempts to approximate $R_m(p)$. Use the same prompt and model settings across these attempts, varying only factors necessary to ensure independence (e.g., internal sampling randomness).

(2) **Estimate per-attempt cost.** Track the average number of tokens (prompt + generation) consumed per attempt, multiply by the current token price (which can differ by model provider or usage level), and add any extra charges (e.g., third-party API calls, external reasoning modules, etc.). This sum yields $C_m(p)$.

(3) **Compute cost-of-pass.** For each model $m$, calculate $v(m, p) = C_m(p)/R_m(p)$. ($R_m(p) = 0$ yields $v(m, p) = \infty$.)

(4) **Determine frontier cost-of-pass.** Estimate human-expert cost $v(\text{expert}, p)$ (see below). Find $V_p(\mathcal{M} \cup \mathcal{M}_0)$ for a given set of strategies $\mathcal{M}$.

(5) **Analyze over benchmarks.** Aggregate $V_p(\mathcal{M})$ across problems $p \sim D$ to estimate $V_{p \sim D}(\mathcal{M}_t)$. Track progress over time (for $\mathcal{M}_t$) and compute gain $G_{p \sim D}$ for new models.

### 2.6.1 Estimating Human-Expert Cost

To estimate $v(\text{expert}, p)$, the plausible cost of obtaining a correct human-expert answer, we systematically determine the required qualifications, appropriate hourly compensation, and average time for a typical problem $p$ per dataset. We determine these quantities based on a hierarchy of evidence by prioritizing the dataset's creation process or associated studies (e.g., reported annotation pay/time (Parrish et al., 2022)). When direct data is absent, we leverage findings from closely related work (Zhang et al., 2024) or infer parameters from the dataset's context (e.g., deriving time-per-problem

from contest rules (Art of Problem Solving, 2023)). Compensation rates are informed by reported study payments (Rein, 2024) or relevant market rates for comparable expertise (e.g., specialized tutoring rates (TutorCruncher, 2025; Wyzant, 2025)).[‡]

| Model Category | Basic Quantitative | | Knowledge Based | | Complex Quantitative | |
|---|---|---|---|---|---|---|
| | 2-Digit Add. | GSM8K | BBQ | GPQA Dia. | MATH 500 | AIME24 |
| *Lightweight Models* | | | | | | |
| Llama-3.1-8B | 4.8e−5 | 0.19 | 2.7e−2 | 18.58 | 3.38 | 15.33 |
| GPT-4o mini | 5.4e−5 | 0.22 | 1.3e−2 | 25.38 | 2.06 | 14.67 |
| Llama-3.3-70B | 1.6e−4 | 0.16 | 7.4e−3 | 18.58 | 1.31 | 10.67 |
| *Large Models* | | | | | | |
| Llama-3.1-405B | 6.9e−4 | 0.14 | 6.7e−3 | 10.43 | 1.13 | 8.67 |
| Claude Sonnet-3.5 | 2.1e−3 | 0.19 | 6.4e−3 | 14.06 | 2.54 | 14.67 |
| GPT-4o | 2.3e−3 | 0.17 | 6.2e−3 | 14.07 | 0.96 | 14.01 |
| *Reasoning Models* | | | | | | |
| OpenAI o1-mini | 5.4e−3 | 0.17 | 1.3e−2 | 12.27 | 0.50 | 4.80 |
| OpenAI o1 | 1.9e−2 | 0.22 | 4.3e−2 | 8.07 | 0.90 | 2.85 |
| DeepSeek-R1 | 1.8e−3 | 0.17 | 1.5e−2 | 14.57 | 0.21 | 3.41 |
| OpenAI o3-mini | 1.1e−3 | 0.11 | 1.1e−2 | 8.18 | 0.76 | 2.03 |

**Table 1:** Frontier dollar cost-of-pass per model / dataset. Each entry is the expected dollar cost of a problem $p \sim D$ with the presence of the model $m$ and a human expert: $V_{p \sim D}(\{m\} \cup \mathcal{M}_0)$. Per column, the 3 entries with the lowest value (i.e. best frontier cost-of-pass) have blue highlights. Different model families emerge as cost-effective at different task categories, highlighting the strengths of our evaluation.

## 3 EXPERIMENTS

### 3.1 EXPERIMENT SETUP

**Models.** We consider three categories of models:

(1) *Lightweight* **models:** We use the per-token cost as a proxy and select models with a cost less than $1 per million input and output tokens (see Table 4): Llama-3.1-8B (Grattafiori et al., 2024), GPT-4o mini (OpenAI, 2024), and Llama-3.3-70B (Meta-AI, 2024).

(2) *Large* **models:** We select large general-purpose LMs: Llama-3.1-405B (Grattafiori et al., 2024), Claude Sonnet-3.5 (Anthropic, 2024), and GPT-4o (Hurst et al., 2024).

(3) *Reasoning* **models:** We select models with special reasoning post-training, including OpenAI's o1-mini (Jaech et al., 2024), o1 (Jaech et al., 2024), and o3-mini (OpenAI, 2025), as well as DeepSeek R1 (Guo et al., 2025).

Within each category, we select three to four representative models released between the second half of 2024 and early 2025. To preserve the integrity of our temporal analysis, we prioritize the earliest stable releases and exclude research previews or experimental versions.

**Datasets.** We evaluate models across three sets of tasks:

(1) *Basic quantitative* **tasks:** These involve basic numerical reasoning. We include an arithmetic dataset (`Two Digit Addition`) to assess basic numerical computation, and GSM8K (Cobbe et al., 2021) to evaluate multi-step grade-school level problem solving.

(2) *Knowledge-based* **tasks:** These require recalling and reasoning over factual knowledge. We include a scientific knowledge-intensive question answering task (`GPQA-Diamond` (Rein et al., 2024)) to evaluate models' ability to recall and utilize complex scientific facts, and a bias benchmark (BBQ (Parrish et al., 2022)) to evaluate whether models rely on stereotypical knowledge or can disambiguate factual responses from biased defaults.

(3) *Complex quantitative reasoning* **tasks:** These require complex mathematical reasoning and problem solving. We use `MATH-500` (Hendrycks et al., 2021; Lightman et al., 2023) to assess models

---

[‡] The full derivation, justification, and sources for our approach are detailed in Appendix A. The resulting estimates are in Table 3.

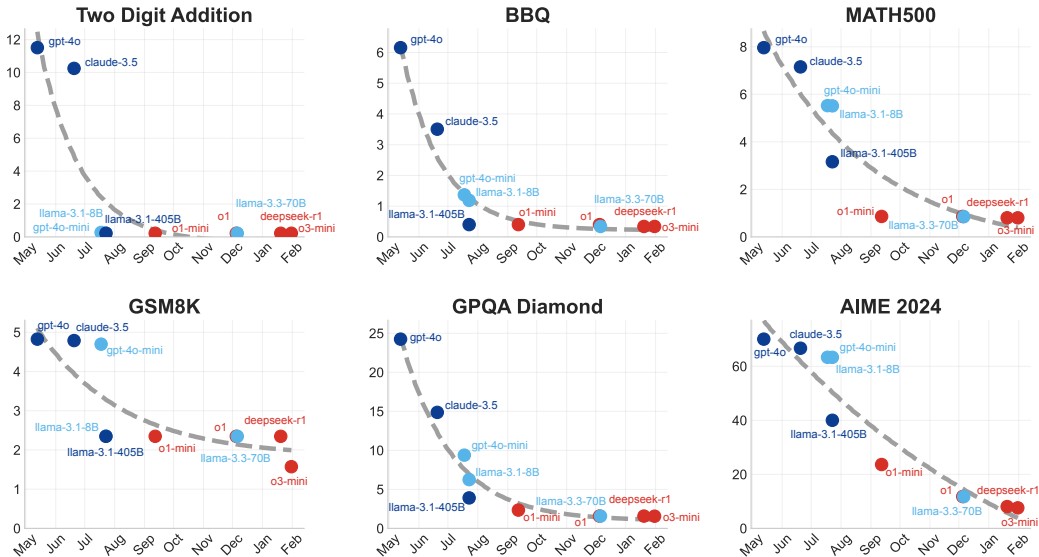

**Figure 2:** The frontier dollar cost-of-pass (i.e. $V_{p \sim D}(\mathcal{M}_t)$) steadily decreases with new model releases, spanning models released between May 2024 and February 2025. Y-axes are normalized (divided by $V_{p \sim D}(\mathcal{M}_0)$, shown in percentage (%)).

on competition-level maths problems, and `AIME-24` (Mathematical Association of America, 2024) to evaluate performance on challenging problems from the 2024 American Invitational Mathematics Examination.

**Evaluation protocol.** All implementation details including model API providers, per-token pricing, prompt template, sampling budget, and accuracy/cost calculation details are shared in Appendix B.

## 3.2 FRONTIER COST-OF-PASS WITH A SINGLE MODEL

In this experiment, we aim to quantify the economic value each model $m$ generates on different distributions of problems $p \sim D$. For this, we take human-expert as a baseline and quantify the frontier cost-of-pass of a problem in the presence of the model $m$: $V_{p \sim D}(\{m\} \cup \mathcal{M}_0)$.

The results in Table 1, highlighting the lowest three instances per dataset, show that our frontier cost-of-pass effectively captures how different model families offer economic advantages across various task categories. We find that lightweight models yield the lowest frontier cost-of-pass on basic quantitative tasks, such as `Two Digit Addition`. This outcome aligns with the observation that all model families achieve high accuracy on this dataset (see Table 5), which in turn makes the least expensive models appear most cost-effective. In contrast, for knowledge-based tasks, larger models achieve a lower frontier cost-of-pass compared to lightweight ones. While the reasoning models, such as o1, are priced significantly more expensively compared to both large and lightweight models, they lead to significant performance improvements, which, overall, result in reductions in the cost-of-pass mainly in complex quantitative tasks.

In contrast, when either task performance ($R_m(p \sim D)$) or cost ($C_m(p \sim D)$) is solely taken into account (Tables 5 and 6) such metrics tend to favor either reasoning or lightweight models respectively due to their significant edge per criteria, without assessing the nuances in the economic impact they induce. This effectively highlights the sophistication of our metric and evaluation framework.

## 3.3 TRACKING FRONTIER COST-OF-PASS WITH NEW RELEASES

In this experiment, we track the improvements on the frontier cost-of-pass for a problem. Figure 2 shows the trends of the cumulative gain per dataset ($V_{p \sim D}(\mathcal{M}_t)$), each updated by the corresponding model release ($\mathcal{M}_{t-1} \cup \{m_t\}$). We observe a steady decline in the frontier cost-of-pass for complex quantitative tasks. In contrast, knowledge-based and basic quantitative tasks typically exhibit a sharp initial drop in frontier cost-of-pass with the early releases of models, followed by a plateau. To

quantify the cost reduction trends, we empirically fit an exponential decay curve of the form:

$$V_p(M_t) \approx a\,e^{-b\,t} + c, \qquad (10)$$

where $t$ denotes time in months since the first model release, and $a$, $b$, and $c$ are fit parameters. From this, we compute the time for the exponential component of the cost to drop by 50%: $T_{1/2} = \ln(2)/b$. Using this formulation, we find that for complex quantitative tasks, between May 2024 and February 2025, the frontier cost-of-pass for `MATH-500` halved approximately every 2.6 months, whereas for `AIME-2024`, the halving time was 7.1 months; indicating consistent cost reductions over the past year.

### 3.4 ESSENTIALNESS OF MODEL FAMILIES: COUNTERFACTUAL FRONTIER COST-OF-PASS

Section 3.3 showed the frontier cost-of-pass decreasing over time with new model releases. To understand which model families were most critical to this progress, we conduct a counterfactual analysis that quantifies the impact of removing each family. Defining $\mathcal{M}_g$ as a family of models (lightweight, large, or reasoning), we measure the counterfactual contribution of family $g$ on dataset $D$ by calculating the relative improvement in frontier cost-of-pass attributable to its inclusion:

$$\frac{G_{p\sim D}(\mathcal{M}_g, \mathcal{M}_T \setminus \mathcal{M}_g)}{V_{p\sim D}(\mathcal{M}_T \setminus \mathcal{M}_g)}. \qquad (11)$$

Here, $\mathcal{M}_T$ includes all models used in our experiments. This metric represents the relative improvement in the final frontier cost-of-pass $V_{p\sim D}(\mathcal{M}_T)$ attributable to the model family $\mathcal{M}_g$, with higher values indicating greater essentialness of that family for achieving the current frontier.

| Model Family Left Out | Basic Quantitative | | Knowledge Based | | Complex Quantitative | |
|---|---|---|---|---|---|---|
| | Two Digit Addition | GSM8K | BBQ | GPQA Diamond | MATH500 | AIME 2024 |
| Lightweight | 93.5 | 50.4 | 75.3 | 33.7 | 2.9 | 0.2 |
| Large | 0.0 | 0.0 | 44.7 | 33.3 | 0.1 | 0.1 |
| Reasoning | 0.0 | 33.0 | 0.0 | 49.9 | 74.4 | 81.0 |

**Figure 3:** The relative improvement (%) in frontier cost-of-pass attributable to each model family $g$, calculated under a counterfactual setting where $\mathcal{M}_g$ is removed. Higher values signify greater essentialness for maintaining the current frontier.

Figure 3 illustrates our main findings, revealing distinct roles across model families. Lightweight models help reduce the frontier cost-of-pass on basic quantitative tasks, while large models are only essential in knowledge-intensive tasks. Reasoning models play a key role in advancing the frontier for complex quantitative reasoning and also improve performance on `GPQA-Diamond`, as well as `GSM8K`, which benefits from small reasoning models like `o3-mini`.

These findings highlight that progress on different task types is driven by different model paradigms. While large models have brought clear gains on knowledge-intensive tasks (e.g. GPQA), improvements in cost-efficiency, especially in more quantitative domains, appear largely driven by advances in lightweight and reasoning models. Together, these suggest that the current cost-efficiency frontier, as reflected in our framework, is shaped mainly by (i) lightweight models and (ii) reasoning models.

### 3.5 IMPACT OF INFERENCE TIME TECHNIQUES ON FRONTIER COST-OF-PASS

We now assess whether common inference-time techniques provide meaningful economic benefits. Specifically, we ask: is it cost-effective to improve model performance through these techniques, compared to relying on the models' baseline performance? To explore this, we focus on the set of lightweight and large models, denoted by $\mathcal{M}_L$. First, we determine the frontier cost-of-pass achieved

by $\mathcal{M}_L$ without any modifications. We then apply a given inference-time technique uniformly across all models in $\mathcal{M}_L$, yielding a modified set $\mathcal{M}_L^*$. The gain from this technique, measured relative to

| Inference Time Technique | Basic Quantitative | | Knowledge Based | | Complex Quantitative | |
|---|---|---|---|---|---|---|
| | Two Digit Addition | GSM8K | BBQ | GPQA Diamond | MATH500 | AIME24 |
| TALE-EP | 1.5 | 66.6 | 24.5 | 50 | 0.2 | 16.6 |
| Self-Refinement | 0 | 0 | 6.7 | 24.9 | 0 | 0 |
| Majority Voting (k=3) | 0 | 0 | 0 | 0 | 0 | 0 |
| Majority Voting (k=4) | 0 | 0 | 0 | 0 | 0 | 0 |

**Table 2:** Relative performance gains (%) from different inference time techniques across datasets.

the original frontier cost-of-pass, can be computed as follows:

$$\frac{G_{p \sim D}(\mathcal{M}_L^*, \ \mathcal{M}_L)}{V_{p \sim D}(\mathcal{M}_L)}. \tag{12}$$

We consider two popular techniques: self-refinement Madaan et al. (2023) and majority voting (a.k.a. self-consistency; Wang et al., 2023), with 3 and 4 votes. Moreover, we evaluate a budget-aware inference-time technique: TALE-EP Han et al. (2025) as well. As shown in Table 2, self-refinement shows some economic benefit on knowledge-intensive tasks, considerably 24.9% improvement on GPQA-Diamond. In contrast, majority voting (despite potentially enhancing accuracy) does not offer relative economic improvement across the tested models and datasets. Meanwhile, the budget-aware technique contributes meaningfully in many more of the tasks to reducing the frontier cost-of-pass.

Collectively, these findings suggest that, for the evaluated techniques, the costs by performance-oriented methods often outweigh accuracy gains when measured by the frontier cost-of-pass. By contrast, TALE-EP (conditioning generation on a self-predicted token budget) yields visible reductions on a subset of tasks, though benefits are uneven. This implies that such common inference-time approaches may currently offer limited economic benefits within our evaluation framework.

## 4 RELATED WORKS

**Economic perspectives and broader impacts.** The efficiency of LMs carries significant economic implications, as they are viewed as general-purpose technologies impacting productivity and labor (Eloundou et al., 2024; Brynjolfsson et al., 2025). Recent scholarship in economics further examines how AI changes which tasks are automated, how expertise is organized, and how R&D productivity evolves across domains (Svanberg et al., 2024; Autor & Thompson, 2025; Besiroglu et al., 2024). Complementary economic analyses explore provider strategies regarding pricing and product design (Bergemann et al., 2025), trade-offs between training and inference compute (Villalobos & Atkinson, 2023), and user-side decision-making involving ROI, token costs, and success probabilities (Xexéo et al., 2024).

Our cost-of-pass metric serves as a bridge between these technical realities of model performance and their economic consequences. By providing the expected monetary cost to successfully complete a task, it allows for quantifying the economic contribution of specific AI systems and informs rational model selection for achieving economic viability, and provides quantitative perspective on the economic evolution of the LM ecosystem.

**LM resource consumption, efficiency optimization and benchmarking.** Research increasingly recognizes the importance of LM resource consumption and efficiency. Empirical studies already connect benchmark performance to monetary cost by tracking LLM inference prices and historical price–performance trends across tasks and benchmarks (Cottier et al., 2025; Gundlach et al., 2025). These empirical evaluations are complemented by work that have quantified operational costs like tokens (Chen et al., 2023) and energy (Maliakel et al., 2025), revealing task-dependent performance and potential diminishing returns from high expenditure (Miserendino et al., 2025). This focus has intensified with the rise of reasoning methodologies (Sui et al., 2025) and inference-time techniques (e.g., Madaan et al. (2023); Wang et al. (2023)), which often trade increased computational cost for potential accuracy gains.

Concerns like "overthinking," where lengthy processing fails to improve results (Chen et al., 2024; Cuadron et al., 2025), have spurred efforts to optimize resource use through methods like dynamic token budgeting (Han et al., 2025), specialized training (Arora & Zanette, 2025), prompt engineering (Xu et al., 2025; Aytes et al., 2025) or researching optimal reasoning lengths (Wu et al., 2025;

Yang et al., 2025). Concurrently, evaluation methodologies have evolved beyond pure accuracy or correctness measures.

Recognizing its insufficiency, researchers have incorporated cost via fixed budgets (Wang et al., 2024), performance heuristics (McDonald et al., 2024), or non-monetary metrics like conciseness (Nayab et al., 2024). Kapoor et al. (2024) strongly advocated for using real dollar costs and accounting for stochasticity—factors central to our approach. Benchmarking efforts have also highlighted diminishing returns from simply scaling inference computation (Parashar et al., 2025). While these works underscore the need for cost-aware analysis, they often rely on specific constraints (e.g., fixed budgets) or heuristic metrics.

Our cost-of-pass framework seeks to advance this by providing a single, interpretable metric that adapts economic production principles, offering a unified way to assess the economic viability of different models and techniques without predefined budget assumptions or proxy metrics.

## 5  CONCLUSION

We introduced an economic framework designed to evaluate language models by integrating their performance with inference cost. Drawing from production theory, we conceptualize language models as stochastic producers, and assess their efficiency using our proposed *cost-of-pass* metric, which measures the expected cost per correct solution. Our analysis utilizes this metric alongside the *frontier cost-of-pass*, defined as the minimum achievable cost compared to a human expert baseline. This approach reveals distinct economic roles played by different model classes. For instance, retrospective and counterfactual evaluations demonstrate that lightweight models primarily drive efficiency on basic tasks, whereas reasoning models are essential for complex problem-solving. Critically, our findings show that common inference-time techniques typically increase the *cost-of-pass*, thus failing to provide net economic benefits when compared to the progress made by improving the underlying models themselves. We discuss the limitations of our methodology, outline directions for future work, and consider practical implications of our framework in Appendix D. Taken together, these insights underscore the value of our framework in offering a principled foundation for measuring language model innovation in economic terms. It serves as a valuable tool for guiding model selection and aligning AI development with real-world value.

## ACKNOWLEDGMENTS

We thank Federico Bianchi, Dan Jurafsky, Daniel E. Ho, Can Yeşildere, Semyon Lomasov and Aaron Scher for valuable comments and discussions in the early stages of this project. MHE gratefully acknowledges support from the Fulbright Foreign Student Program. BE gratefully acknowledges the support of the Stanford Knight-Hennessy Scholarship. MS gratefully acknowledges the support of an HAI-SAP Fellowship and a Google PhD Fellowship.

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

## A   DETAILS OF HUMAN EXPERT COST ESTIMATION

In this section, we provide a detailed analysis of how the human expert costs in Table 3 are calculated per dataset.

| Dataset | Qualification Requirements | Hourly Rate | Time per Question | Est. Cost |
|---|---|---|---|---|
| AIME | Advanced high-school contest math skills | $45–$100 | ~12 minutes | $9–$20 |
| BBQ | General familiarity with social biases | $15 | ~0.4 minutes (24 sec) | $0.10 |
| GPQA Dia. | Graduate-level domain expertise | $100 | ~35 minutes | $58 |
| GSM8K | Basic arithmetic reasoning | $33–$55 | ~3.7 minutes | $2–$3.50 |
| MATH500 | Strong competition-level problem-solving | $35–$60 | ~12 minutes | $7–$12 |
| Two-Digit Add. | Basic numeracy | $10–$20 | ~0.04 minutes (~2-3 sec) | $0.01–$0.02 |

**Table 3:** Estimated costs of hiring a human expert to solve one question from each dataset, based on typical qualifications, hourly rates, and time per question.

**AIME** (American Invitational Mathematics Examination) consists of 15 challenging math problems in a 3-hour contest (administered in two separate sections: AIME I & II), giving an average of about 12 minutes per problem (Art of Problem Solving, 2023). In practice, expert math tutors for competitions like AIME command high hourly fees in the range of $45–$100, reflecting intensive test-preparation rates (TutorCruncher, 2025). This rate range aligns with specialized test prep tutoring in the US, which is higher than regular tutoring due to the advanced problem-solving skills required (TutorCruncher, 2025). At roughly 12 minutes per AIME question on average, a solver could handle about five such problems per hour under exam conditions (Art of Problem Solving, 2023).

**BBQ** (Bias Benchmark for QA) contains short question-answer scenarios targeting social bias. Crowdworkers annotating BBQ have been paid around $15 per hour, a rate chosen to exceed U.S. minimum wage (Parrish et al., 2022). Because each task includes multiple BBQ questions, workers were able to answer roughly 5 questions in 2 minutes (Parrish et al., 2022) – i.e. ~**24 seconds per question**, or about 0.4 minutes per question. This fast per-question time reflects the fact that BBQ items are short multiple-choice queries, allowing a human annotator to complete approximately 150 BBQ questions in an hour at that pay rate (Parrish et al., 2022).

**GPQA-Diamond** consists of extremely difficult graduate-level science questions, so human experts demand high compensation. In one case, domain experts were paid about $100 per hour to contribute and validate GPQA questions (Rein et al., 2024). These questions are "Google-proof" and time-consuming: skilled non-expert participants spent **over 30–35 minutes on average per question** when attempting to solve GPQA problems with unrestricted web access (Rein et al., 2024). This long duration per question underscores GPQA's complexity: at most 2 questions could be solved in an hour even by motivated annotators, which justifies the premium expert hourly rate (Rein, 2024).

**GSM8K** contains grade-school level math problems. Solving these is relatively time-efficient for adults: in one study, crowdworkers under time pressure managed to solve about 4.07 GSM8K problems in 15 minutes on average (Zhang et al., 2024), or roughly **3.7 minutes per question**. The required skill is comparable to general math tutoring at the K-8 level, for which typical U.S. tutor rates are about $33–$55 per hour on platforms like Wyzant (Wyzant, 2025). At such a rate, paying a person to solve GSM8K problems would be economical, given that a proficient solver can complete approximately 16 questions in one hour (Zhang et al., 2024).

**MATH-500** is a set of 500 advanced competition math problems (drawn from the larger MATH dataset). These span a range of difficulty levels (from relatively routine competition questions up to AIME-level problems) so a strong general competition-math tutor has typically sufficient qualifications. As with AIME, a well-prepared human might spend on the order of 10–15 minutes per problem, roughly ~**12 minutes on average for a hard competition question** (Art of Problem Solving, 2023). Tutors capable of solving and teaching such competition math problems often charge rates on the order of $50 per hour (with a typical range of $35–$60 for competition math tutoring)

(Wyzant, 2025). Therefore, solving roughly five MATH-500 problems could cost about \$50 and take around an hour, consistent with the per-question time and high skill required.

**Two-Digit Addition** consists of simple two-digit addition problems, which are very quick for humans to solve. Early elementary school students are often expected to complete about 20-30 basic addition problems in one minute (Kirchner, 2017; Rocket Math, 2022). This corresponds to roughly **2–3 seconds per addition** ($\sim$0.04 minutes per question). Because the task is so elementary, the labor to solve large numbers of such problems can be valued at a lower hourly rate. Simple data-entry style work or basic math tasks on freelance platforms pay on the order of \$10–\$20 per hour (Upwork, 2025a). At \$15/hour, for example, a worker could theoretically solve several hundred 2-digit additions within the hour, given the $\sim$2-3 second solution time (Kirchner, 2017; Rocket Math, 2022).

## B  DETAILS OF EVALUATION

For each dataset in our evaluation, we sample up to $128$ instances and run each model[§] $n = 8$ times to estimate the expected runtime cost and accuracy per sample. We use a temperature of $0.7$ and top_p of $1.0$ for all models except OpenAI's reasoning models, for which we set the temperature to $1.0$ without applying top_p. Additionally, we use the default maximum token generation limits provided by each model. Following Suzgun et al. (2025), we use a concise but descriptive instruction prompt for models to follow:

```
Experiment Prompt

Please solve the following question.  You can explain your solution before
presenting the final answer. Format your final answer as:

<answer>
...
</answer>

Instructions:
- For multiple-choice: Give only the letter (e.g., (A)).
- For numeric: Give only the number (e.g., 42).
- For free-response: Provide the full final answer text.

INPUT:

'''
{input}

'''
```

In our experiments, we define the pass $r_m(p)$ as whether the model obtains a correct answer after a single run or not (0 or 1), and the cost $c_m(p)$ as:

$$c_m(p) = n_{\text{in}}(m, p) \cdot c_{\text{in}}(m) + n_{\text{out}}(m, p) \cdot c_{\text{out}}(m) \tag{13}$$

where $n_*(m, p)$ denotes the number of input / output tokens consumed / generated by the model $m$ on problem $p$, and $c_*(m)$ denotes the dollar costs per input / output tokens consumed / generated by the model $m$ (see Table 4 for the pricing). For the expert costs, we utilize the estimations from Table 3, and set the rates to the upper-bound value to ensure the approximation of the expert accuracy being 1. Finally, as shown in Table 4, we access proprietary models via their original providers, while open-source models are queried through a single provider for consistency and simplicity (TogetherAI, in our case).

---

[§] Here, the short-form "model" refers to the underlying model together with its inference pipeline (prompt, decoding settings, etc.). Comparisons throughout the paper are done on this basis, and Appendix B shares the adopted details.

| Category | Model | Release Date | Cost (per million tokens) | |
|---|---|---|---|---|
| | | | Input Tokens | Output Tokens |
| Lightweight Models | Llama-3.1-8B | 7/23/2024 | $0.18 | $0.18 |
| | GPT-4o Mini | 7/18/2024 | $0.15 | $0.60 |
| | Llama-3.3-70B | 12/6/2024 | $0.88 | $0.88 |
| Large Models | Llama-3.1-405B | 7/23/2024 | $3.50 | $3.50 |
| | GPT-4o | 5/13/2024 | $2.50 | $10.00 |
| | Claude Sonnet-3.5 | 6/20/2024 | $3.00 | $15.00 |
| Reasoning Models | OpenAI o1-mini | 9/12/2024 | $1.10 | $4.40 |
| | OpenAI o3-mini | 1/31/2025 | $1.10 | $4.40 |
| | DeepSeek-R1 | 1/20/2025 | $7.00 | $7.00 |
| | OpenAI o1 | 12/5/2024 | $15.00 | $60.00 |

**Table 4:** Per-token inference costs with release dates. Each model name links to the utilized provider.

## C  ADDITIONAL RESULTS

### C.1  EXPECTED ACCURACY AND INFERENCE COSTS

As discussed in Section 3.2, we report the expected accuracy and cost for each model per dataset, denoted as $R_m(p \sim D)$ and $C_m(p \sim D)$. To compute these, following the methodology in Section 2.5, we use the i.i.d. sampled set $P \sim D$ of problems per dataset and approximate the expectation by averaging the accuracy $R_m(p)$ and cost $C_m(p)$ across problem instances. The results in Tables 5 and 6 reveal a skewed preference for particular model families under each metric, suggesting that these metrics alone are insufficient to capture the economic impact of models.

| Model Category | Basic Quantitative | | Knowledge Based | | Complex Quantitative | |
|---|---|---|---|---|---|---|
| | 2-Digit Add. | GSM8K | BBQ | GPQA Dia. | MATH 500 | AIME24 |
| *Lightweight Models* | | | | | | |
| Llama-3.1-8B | 89.45 | 75.78 | 21.48 | 17.87 | 37.30 | 12.50 |
| GPT-4o mini | 99.90 | 88.57 | 53.32 | 18.07 | 70.02 | 14.58 |
| Llama-3.3-70B | 99.90 | 92.09 | 85.06 | 46.48 | 72.75 | 33.33 |
| *Large Models* | | | | | | |
| Llama-3.1-405B | 99.71 | 93.95 | 85.74 | 44.14 | 67.87 | 31.67 |
| Claude Sonnet-3.5 | 100 | 94.43 | 92.58 | 55.37 | 64.75 | 15.83 |
| GPT-4o | 99.71 | 91.99 | 90.04 | 47.07 | 73.14 | 14.58 |
| *Reasoning Models* | | | | | | |
| OpenAI o1-mini | 99.51 | 92.58 | 85.74 | 49.12 | 85.94 | 53.33 |
| OpenAI o1 | 100 | 94.04 | 95.02 | 73.83 | 89.45 | 72.50 |
| DeepSeek-R1 | 100 | 93.36 | 83.69 | 54.88 | 93.85 | 60.83 |
| OpenAI o3-mini | 100 | 92.77 | 83.79 | 71.68 | 88.57 | 77.08 |

**Table 5:** Accuracy (%) per model per dataset: $R_m(p \sim D)$. In each column, the 3 entries with the highest accuracy have blue highlights.

### C.2  EVALUATION ON A REAL-WORLD DOMAIN

We evaluate our framework on Tau-bench (Yao et al., 2024), a benchmark that targets tool use, agent behavior, and user interaction in real-world domains. We sample 8 tasks per category (airline, retail), totaling 16 tasks, and run each model as an agent under the evaluation protocol described in the original paper. We exclude DeepSeek-R1 due to its visible chain-of-thought being mixed with user messages, which contaminates responses under this protocol. We apply the cost modeling based on total tokens consumed or generated per round, and we aggregate costs over interaction rounds. Estimates are averaged over 4 independent trials per run.

For the human-expert baseline, we consider the "retail or call-center communication" qualification, with an hourly wage of $20.59 (U.S. Bureau of Labor Statistics, 2025) and an average of 6 minutes per task (Zendesk, 2025), which yields $2.06 per task.

| Model Category | Basic Quantitative | | Knowledge Based | | Complex Quantitative | |
|---|---|---|---|---|---|---|
| | 2-Digit Add. | GSM8K | BBQ | GPQA Dia. | MATH 500 | AIME24 |
| *Lightweight Models* | | | | | | |
| Llama-3.1-8B | 4.2e−5 | 7.4e−5 | 5.2e−5 | 1.8e−4 | 1.5e−4 | 2.2e−4 |
| GPT-4o mini | 5.4e−5 | 1.9e−4 | 1.0e−4 | 3.9e−4 | 3.7e−4 | 5.6e−4 |
| Llama-3.3-70B | 1.6e−4 | 3.3e−4 | 3.1e−4 | 9.6e−4 | 6.7e−4 | 1.1e−3 |
| *Large Models* | | | | | | |
| Llama-3.1-405B | 6.9e−4 | 1.4e−3 | 1.0e−3 | 3.0e−3 | 2.4e−3 | 3.7e−3 |
| Claude Sonnet-3.5 | 2.1e−3 | 3.7e−3 | 3.0e−3 | 6.9e−3 | 5.9e−3 | 7.5e−3 |
| GPT-4o | 2.3e−3 | 4.5e−3 | 2.7e−3 | 0.01 | 8.7e−3 | 0.01 |
| *Reasoning Models* | | | | | | |
| OpenAI o1-mini | 5.4e−3 | 8.4e−3 | 7.6e−3 | 0.02 | 0.02 | 0.07 |
| OpenAI o1 | 0.02 | 0.03 | 0.04 | 0.25 | 0.13 | 0.52 |
| DeepSeek-R1 | 1.8e−3 | 5.1e−3 | 4.6e−3 | 0.04 | 0.01 | 0.04 |
| OpenAI o3-mini | 1.1e−3 | 2.1e−3 | 2.6e−3 | 0.01 | 5.4e−3 | 0.02 |

**Table 6:** Dollar cost incurred per model per dataset: $C_m(p \sim D)$. In each column, the 3 entries with the lowest cost have blue highlights.

| *Lightweight Models* | | *Large Models* | | *Reasoning Models* | |
|---|---|---|---|---|---|
| Llama-3.1-8B | 1.6770 | Llama-3.1-405B | 1.8875 | OpenAI o1-mini | 1.8230 |
| GPT-4o mini | 1.2944 | Claude Sonnet-3.5 | 1.5135 | OpenAI o1 | 1.6406 |
| Llama-3.3-70B | 1.6897 | GPT-4o | 1.2247 | OpenAI o3-mini | 1.2703 |

**Table 7:** Frontier dollar cost-of-pass per model on `Tau-bench` real-world tasks. Each pair of columns lists models (left) and their frontier cost-of-pass with respect to the human expert baseline (right): $V_{p \sim D}(\{m\} \cup \mathcal{M}_0)$. The lowest three values are highlighted in blue, indicating that all the model families have an economic merit in this task.

We repeat the analyses in Sections 3.2, 3.3, and 3.4; and share the results in Tables 7, 8, 9 respectively. Our overall findings indicate that (1) all model families have a merit in this task, (2) the evolution of the frontier cost-of-pass still follows an exponential decay (similar to other tasks), and (3) none of the model families are significantly essential in driving progress.

## C.3 RELATIVE GAIN PER MODEL RELEASE

Figure 4 presents the relative improvement in temporal frontier cost-of-pass for each model release, illustrated using bar plots. Namely, we calculate:

$$\frac{G_{p \sim D}(\{m_t\}, \mathcal{M}_{t-1})}{V_{p \sim D}(\mathcal{M}_{t-1})} \tag{14}$$

The results indicate that the reasoning models demonstrate notable advancements, particularly on complex quantitative tasks. In contrast, lightweight models exhibit marked gains on basic tasks. These findings support the observations from our experiments (Sections 3.2, 3.4). Notably, The substantial improvement observed for GPT-4o is likely due to it being the first model included in our analysis, resulting in a pronounced leap relative to the baseline cost associated with human expert annotation.

| May 13 | Jun 20 | Jul 18 | Jul 23 | Sep 12 | Dec 5 | Dec 6 | Jan 31 |
|---|---|---|---|---|---|---|---|
| | Claude | GPT-4o | Llama-3.1-8B | OpenAI | Llama-3.3 | OpenAI | OpenAI |
| GPT-4o | Sonnet-3.5 | mini | Llama-3.1-405B | o1-mini | 70B | o1 | o3-mini |
| 1.2247 | 1.1900 | 0.8411 | 0.8127 | 0.8127 | 0.8021 | 0.7668 | 0.7311 |

**Table 8:** Frontier dollar cost-of-pass over model release dates on `Tau-bench`. Each column reports the best-to-date frontier value $V_{p \sim D}(\mathcal{M}_t)$ after incorporating models released on the indicated date. The trajectory continues to follow an exponential decay, consistent with other tasks. This table is the tabular version of the time-evolution figure (see Fig. 2 for example).

|  | Lightweight | Large | Reasoning |
|---|---|---|---|
| Essentialness (%) | 22.5 | 13.2 | 6.0 |

**Table 9:** Essentialness of model families on `Tau-bench` (metric from Section 3.4). The results show that Lightweight models are the most essential, but overall, none of the families are strongly essential in driving progress for the frontier cost-of-pass.

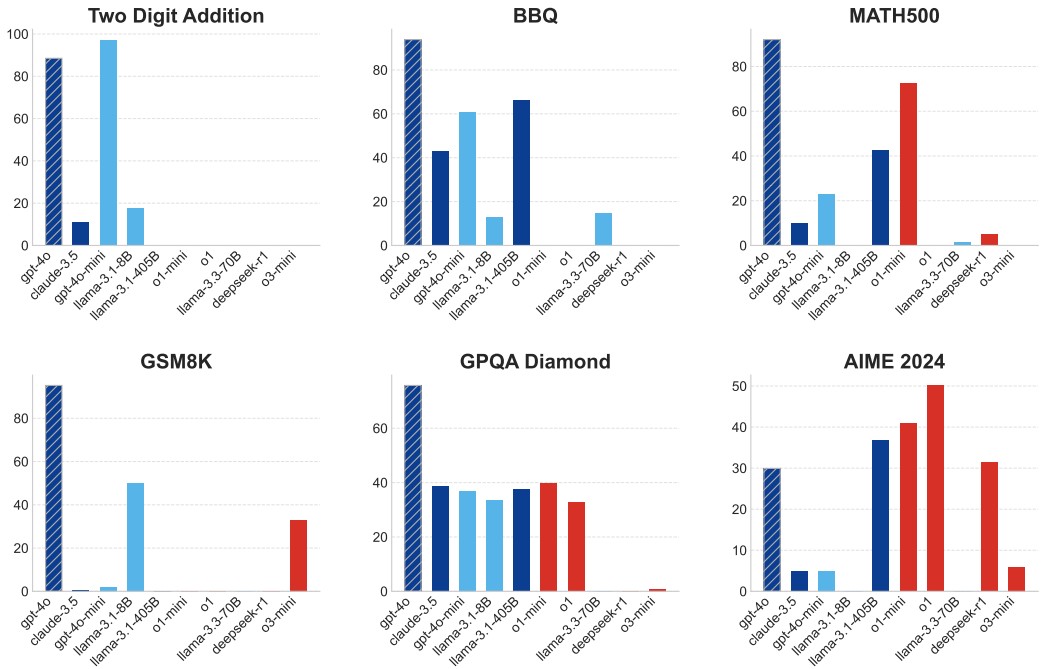

**Figure 4:** Bar plot showing the percentage of change in frontier cost-of-pass per model release (i.e. $\frac{G_{p \sim D}(\{m_t\}, \mathcal{M}_{t-1})}{V_{p \sim D}(\mathcal{M}_{t-1})}$)

## C.4    ESSENTIALNESS OF HUMAN EXPERT BASELINE

Adapting the methodology in Section 3.4, we quantify the essentialness of the human-expert baseline for each task. We treat the human-expert baseline as a separate family, $\mathcal{M}_0$, and compare it to the remaining models, $\mathcal{M}_T \setminus \mathcal{M}_0$, via

$$\frac{G_{p \sim D}(\mathcal{M}_0, \mathcal{M}_T \setminus \mathcal{M}_0)}{V_{p \sim D}(\mathcal{M}_T \setminus \mathcal{M}_0)}. \tag{15}$$

Under this definition, essentialness is $100\%$ if there exists at least one instance in the distribution that no model in $\mathcal{M}_T \setminus \mathcal{M}_0$ can solve, so the frontier requires $\mathcal{M}_0$ on some part of the distribution. Conversely, if every instance is solved at strictly lower cost by the LMs, essentialness is $0\%$.

Applying this analysis, we find that human experts remain fully essential for `GSM8K`, `GPQA-Diamond`, `MATH-500`, and `AIME-2024`, and non-essential ($0\%$) for `BBQ` and `Two-Digit Addition`. Interestingly, there is no task where human experts are partially necessary (in between 0-100%).

## C.5    ESSENTIALNESS OF SINGLE MODELS

In this section, following the methodology outlined in Section 3.4, we quantify the relative improvement in frontier cost-of-pass using a counterfactual approach. Specifically, for each model $m_*$, we calculate the following:

$$\frac{G_{p \sim D}(\{m_*\}, \mathcal{M}_T \setminus \{m_*\})}{V_{p \sim D}(\mathcal{M}_T \setminus \{m_*\})}, \tag{16}$$

quantifying the essentialness of the model $m_*$. The results presented in Figure 5 demonstrate that the contributions of most individual models are largely compensable by the remaining models.

Furthermore, we observe a similar coarse-level trend, as noted in Section 3.4, indicating that different model families provide greater benefits in specific task categories.

| | | Basic Quantitative | | Knowledge Based | | Complex Quantitative | |
|---|---|---|---|---|---|---|---|
| | | TwoDigitAddition | GSM8K | BBQ | GPQA Diamond | MATH500 | AIME 2024 |
| Lightweight | llama-3.1-8B | 17.8 | 49.9 | 13.5 | 0.3 | 0.2 | 0.0 |
| | gpt-4o-mini | 7.2 | 0.0 | 32.4 | 0.1 | 0.2 | 0.0 |
| | llama-3.3-70B | 0.0 | 0.0 | 14.9 | 0.1 | 0.5 | 0.1 |
| Large | llama-3.1-405B | 0.0 | 0.0 | 34.7 | 33.3 | 0.1 | 0.1 |
| | claude-3.5 | 0.0 | 0.0 | 0.0 | 0.0 | 0.0 | 0.0 |
| | gpt-4o | 0.0 | 0.0 | 6.9 | 0.0 | 0.0 | 0.0 |
| | o1-mini | 0.0 | 0.0 | 0.0 | 0.0 | 0.0 | 0.0 |
| Reasoning | o1 | 0.0 | 0.0 | 0.0 | 0.0 | 0.0 | 24.8 |
| | deepseek-r1 | 0.0 | 0.0 | 0.0 | 0.0 | 4.2 | 0.0 |
| | o3-mini | 0.0 | 33.0 | 0.0 | 0.8 | 0.3 | 5.9 |

(Left axis label: **Model Left Out**)

**Figure 5:** The relative improvement (%) in frontier cost-of-pass under a counterfactual setting, removing a model $m_*$ from the model set $\mathcal{M}_T$. High values mean that the model is essential for maintaining the current frontier.

## C.6 REGIONAL ABLATION FOR HUMAN EXPERT COST ESTIMATION

As a realistic sensitivity analysis of the human-expert cost estimates, we change the region to India and re-estimate the hourly rates for the stated qualifications in Table 3. The estimates in Table 10 report the updated hourly rates and the resulting estimated cost per question for all datasets. We then repeat the analysis in Section 3.2 and present the results in Table 11. Our findings indicate that while the main interpretation of the analysis does not change, there is a slight shift in preference toward lightweight models when the region is set to India. We interpret this as an effect of the human-expert cost influencing the penalty for failure: when the human-expert cost is lower, the framework becomes more forgiving of errors, thereby increasing the relative appeal of lightweight models.

| Dataset | Hourly Rate (INR) | Reference | Est. Cost (INR) | Est. Cost (USD) |
|---|---|---|---|---|
| AIME | 1,250–2,000 | (TeacherOn, 2025a) | 250–400 | $2.84–$4.55 |
| BBQ | 445 | (SalaryExpert, 2025) | 2.97 | $0.034 |
| GPQA Dia. | 3,520–4,440 | (Upwork, 2025b) | 2,053–2,567 | $23.33–$29.17 |
| GSM8K | 350–800 | (TeacherOn, 2025b) | 21.59–49.33 | $0.25–$0.56 |
| MATH500 | 600–1,200 | (TeacherOn, 2025c) | 120–240 | $1.36–$2.73 |
| Two-Digit Add. | 175–275 | (InputiX, 2020) | 0.12—0.18 | $0.0013–$0.0021 |

**Table 10:** Human-expert cost estimation when the region is changed to India. The qualification requirements and time per question are used from Table 3. The conversion rate of INR88:$1 is used.

## C.7 CONFIDENCE INTERVALS FOR FRONTIER COST-OF-PASS ESTIMATES

To test statistical significance of our results in Section 3.2 and Table 1, we calculate 95% bootstrap percentile confidence intervals for each frontier cost-of-pass estimate in the table by simulating the sampling procedure from the models 10,000 times. By considering the midpoints of the confidence intervals as new estimates, we conduct the same analysis in the main paper. The analysis in Table 12 mostly mirror the original findings with a minor difference that on GPQA Diamond, o1-mini achieves a

| Model Category | Basic Quantitative | | Knowledge Based | | Complex Quantitative | |
|---|---|---|---|---|---|---|
| | 2-Digit Add. | GSM8K | BBQ | GPQA Dia. | MATH 500 | AIME24 |
| *Lightweight Models* | | | | | | |
| Llama-3.1-8B | 4.8e−5 | 0.031 | 0.009 | 9.34 | 0.768 | 3.49 |
| GPT-4o mini | 5.4e−5 | 0.035 | 0.005 | 12.76 | 0.47 | 3.34 |
| Llama-3.3-70B | 1.6e−4 | 0.027 | 0.003 | 9.35 | 0.30 | 2.43 |
| *Large Models* | | | | | | |
| Llama-3.1-405B | 6.9e−4 | 0.023 | 0.003 | 5.25 | 0.261 | 1.98 |
| Claude Sonnet-3.5 | 0.002 | 0.034 | 0.004 | 7.08 | 0.585 | 3.34 |
| GPT-4o | 0.002 | 0.031 | 0.005 | 7.09 | 0.231 | 3.2 |
| *Reasoning Models* | | | | | | |
| OpenAI o1-mini | 0.002 | 0.035 | 0.01 | 6.19 | 0.137 | 1.19 |
| OpenAI o1 | 0.002 | 0.061 | 0.032 | 4.24 | 0.313 | 1.29 |
| DeepSeek-R1 | 0.002 | 0.033 | 0.009 | 7.36 | 0.062 | 0.836 |
| OpenAI o3-mini | 0.001 | 0.02 | 0.006 | 4.13 | 0.178 | 0.488 |

**Table 11:** Using the same experimental setup as in Table 1, except with the region set to India (for which we re-estimate the human-expert baseline costs in Table 10). The results majorly mirror those in Table 1, with a slight shift toward our metric favoring lightweight models.

lower frontier cost-of-pass than Llama-3.1-405B. Therefore, this indicates the statistical significance of our findings.

| Model Category | Basic Quantitative | | Knowledge Based | | Complex Quantitative | |
|---|---|---|---|---|---|---|
| | 2-Digit Add. | GSM8K | BBQ | GPQA Dia. | MATH 500 | AIME24 |
| *Lightweight Models* | | | | | | |
| Llama-3.1-8B | (4.8e−5, 2.0e−4) | (0.192, 0.328) | (0.034, 0.042) | (23.11, 28.09) | (3.84, 4.59) | (15.33, 16.67) |
| GPT-4o mini | (5.3e−5, 5.5e−5) | (0.219, 0.274) | (0.013, 0.017) | (28.55, 32.63) | (2.06, 2.34) | (14.67, 16) |
| Llama-3.3-70B | (1.6e−4, 1.6e−4) | (0.164, 0.192) | (7.4e−3, 9.7e−3) | (19.49, 22.2) | (1.41, 1.78) | (10.67, 12) |
| *Large Models* | | | | | | |
| Llama-3.1-405B | (6.9e−4, 7.0e−4) | (0.138, 0.165) | (6.7e−3, 8.3e−3) | (12.69, 16.77) | (1.32, 1.88) | (8.67, 10.67) |
| Claude Sonnet-3.5 | (2.1e−3, 2.2e−3) | (0.195, 0.195) | (6.3e−3, 7.4e−3) | (14.96, 17.68) | (2.63, 3.1) | (14.67, 15.34) |
| GPT-4o | (2.3e−3, 2.3e−3) | (0.169, 0.196) | (6.1e−3, 8.7e−3) | (15.42, 18.14) | (1.14, 1.61) | (14.01, 16.01) |
| *Reasoning Models* | | | | | | |
| OpenAI o1-mini | (5.3e−3, 5.5e−3) | (0.172, 0.2) | (0.013, 0.014) | (12.73, 15.44) | (0.497, 0.779) | (4.77, 6.1) |
| OpenAI o1 | (0.018, 0.018) | (0.221, 0.223) | (0.042, 0.044) | (8.08, 9.85) | (0.885, 0.988) | (2.75, 3.98) |
| DeepSeek-R1 | (1.8e−3, 1.8e−3) | (0.17, 0.2) | (0.014, 0.017) | (15.46, 17.73) | (0.205, 0.394) | (3.4, 4.74) |
| OpenAI o3-mini | (1.1e−3, 1.1e−3) | (0.112, 0.167) | (0.011, 0.014) | (8.63, 10.89) | (0.756, 0.944) | (2.03, 2.7) |

**Table 12:** The same experimental setup as in Table 1, where we report 95% bootstrap percentile confidence intervals (10,000 bootstrap samples). For each column, the three entries with the lowest midpoint (i.e., best frontier cost-of-pass) are highlighted in blue. The results closely mirror those in Table 1, except that on GPQA Diamond o1-mini achieves a lower frontier cost-of-pass than Llama-3.1-405B; indicating statistically significant results.

## C.8 BENCHMARK HACKING: CHEAP RANDOM GUESSERS ON FORGIVING MULTIPLE-CHOICE TASKS

Our framework is designed to reward strategies that convert monetary cost into correct outputs efficiently. However, in sufficiently forgiving settings its canonical form has a structural weak spot that an extremely cheap yet essentially random agent can appear economically optimal, despite being useless (or even malicious) from an application perspective.

Consider a dataset composed of multiple-choice questions with $k$ options each (e.g. $k = 4$). We assume a model that has just enough capability to follow formatting instructions (e.g. always returning a single option token such as "A", "B", "C", or "D"), but whose behavior on the task is indistinguishable from uniform random guessing, which would lead to $R_{m_{\text{rand}}}(p) \approx \frac{1}{k}$. Its cost-of-pass would then be

$$v(m_{\text{rand}}, p) = \frac{C_{m_{\text{rand}}}(p)}{R_{m_{\text{rand}}}(p)} \approx k \cdot C_{m_{\text{rand}}}(p).$$

When we consider $C_{m_{\text{rand}}}(p)$ to be significantly small (for example, for a low-price model consuming a small number of tokens and emitting $O(1)$ answer tokens), then $v(m_{\text{rand}}, p)$ can be lower than the

cost-of-pass of more capable models that actually reason about the problem but incur higher token costs and/or higher prices per token.

| Dataset | Lowest 3 frontier cost-of-pass | | | Adversarial baseline |
|---------|-----|-----|-----|----------------------|
|         | 3rd | 2nd | 1st |                      |
| BBQ     | 6.7e$-$3 | 6.4e$-$3 | 6.2e$-$3 | **7.9e$-$5** |
| GPQA-Diamond | 10.43 | 8.18 | 8.07 | **2.1e$-$4** |

**Table 13:** Comparison of the three lowest frontier dollar cost-of-pass values per dataset from Table 1 with an adversarial random-guessing baseline (Llama-3.1-8B configured as $m_{\text{rand}}$) on the multiple-choice benchmarks (BBQ, GPQA-Diamond). The adversarial baseline attains an artificially low cost-of-pass, illustrating how a cheap random guesser can exploit forgiving multiple-choice evaluations.

To make the example concrete, we instantiate $m_{\text{rand}}$ directly with Llama-3.1-8B's setup. We configure the model to only output a random answer in the required format by following the instructions in the prompt in Section B. We then repeat the same analysis as in Section 3.2 on the multiple-choice tasks in our framework: BBQ and GPQA-Diamond. Results in Table 13 show that this adversarial baseline attains a substantially lower frontier cost-of-pass than any other reported model, i.e., it appears to have made significantly more progress on top of the human-expert baseline than the genuinely competent systems. This concretely illustrates the benchmark-hacking concern we formalized above.

To mitigate such benchmark hacking in practice, we recommend (i) explicitly excluding impractical baselines such as random guessers or unreliable systems from the set of strategies considered on the frontier, (ii) employing robustness-oriented dataset augmentation that makes random guessing unstable (for example by evaluating each question under several independently shuffled option orderings and requiring the model to produce a consistent semantic answer across these variants) (iii) using stricter success criteria for a pass such as pass^k, which exponentially penalize low-accuracy models, and (iv) incorporating explicit penalties for failed attempts into the cost model. Some of these extensions are elaborated in Section D.1.

# D PRACTICAL IMPLICATIONS, LIMITATIONS, AND FUTURE DIRECTIONS

In this section, we acknowledge the limitations of our framework and evaluations, share practical perspectives together with directions for future extensions.

## D.1 EXTENDING MODELING ACROSS COMPLEX AND DIVERSE DIMENSIONS

Our experiments consider a common but relatively simple cost and performance modeling, which may not seem clear for practitioners to adapt to their more complex settings. To start with, our analyses use per-token API prices that can be represented by $C_m(p) = \mathbf{w}^\top \mathbf{x}_m(p)$ (Section 2.2), where $\mathbf{w}$ contains prices (input/output tokens) and $\mathbf{x}_m(p)$ contains the corresponding quantities. In practical scenarios, one may include other components of the evaluation pipeline by placing their *unit cost* in $\mathbf{w}$ and their *per-attempt quantity* in $\mathbf{x}$ to enrich the definition. Examples include: verification costs per attempt (e.g. human or automatic checks), costs associated with unsuccessful outputs, tool-usage fees, orchestration overhead (e.g. queue time, cold-start penalties, inter-service latency), and amortized fixed costs per attempt (training, hardware depreciation, maintenance, KV caching). Additionally, depending on the nature of the model inference, the costs associated with input and output token consumption can be adjusted (for example, to reflect cost reductions from batched inputs or outputs).

Regarding the success metric, one may replace accuracy with a stricter reliability-oriented metric (e.g., pass^k (Yao et al., 2024), requiring $k$ consecutive successes) or a more lenient metric (e.g., pass@k (Chen et al., 2021), rewarding any success within $k$ attempts). Such alternatives are useful in settings where consistency, robustness, or partial correctness matter. Similarly, for classification-based tasks where precision–recall trade-offs are central, one can treat the desired base criteria (e.g. enforcing a minimum recall at an acceptable false-positive rate) as the definition of a pass and compute cost-of-pass accordingly.

To clearly illustrate how such extensions could be incorporated, let's take our canonical evaluation on Tau-Bench in Section C.2 and imagine its extension to a real-world customer support center that automates part of its customer support workflow with LMs. While our evaluation already

incorporates multiple turns (by accounting for all consumed and generated tokens across rounds) and thus improves on single input/output evaluations, it can still be extended to better represent economic objectives. For instance, one could assign a cost to each tool invocation, capturing both the direct cost of using the tool and a monetary representation of the time it takes to use it. Moreover, one might incorporate verification by a qualified expert who reviews the interaction history and fixes problems, with the expert's labor cost included in the cost formulation. For failures where the expert either manually provides a fix or fails to identify/fix the issue, the cost model could be extended to include the associated losses (e.g. an extra charge for non-catastrophic corrections, or larger losses due to catastrophic errors affecting the customer, such as misbooking an airline ticket). If such a system handles queries in batches and uses a common system prompt, these effects can be incorporated by adjusting the per-token costs and amortizing the system prompt cost across the batch. Finally, other amortized costs from the underlying system (such as maintenance of any infrastructure) can be incorporated into the cost modeling on a per-response basis. Regarding performance, one could apply the reliability-oriented pass^k metric to better and more reliably represent success in real-world deployments when benchmarking different systems or models.

For some applications, alternative units per attempt (FLOPs, time, latency, energy) may matter more than dollar cost, and the application of our framework may not be immediately visible. If an oracle system $m'$ guarantees a non-zero success rate $R_{m'}(p \sim D)$ with a measurable expenditure, one may treat it as a baseline (analogous to the human expert baseline for costs) and apply our analyses, yielding an alternative unit to dollars. If such an oracle does not exist, assuming that for each $p \sim D$ there exists some $m \in \mathcal{M}$ with $R_m(p) > 0$, several analyses in this paper (e.g. essentialness and impact) still apply. In such alternative units, the relative economic desirability of different techniques can also change. For instance, if latency is the main expenditure of interest, some inference-time techniques like majority voting (Wang et al., 2023) can become very effective when the votes are executed in parallel, benefiting from improved expected correctness while incurring roughly the latency of a single vote.

## D.2 INTERPRETABILITY OF OUR FRAMEWORK

Since our cost-of-pass metric is $v(m, p) = \frac{C_m(p)}{R_m(p)}$ for a given problem instance $p$ and model $m$, with $C_m(p) = \mathbf{w}^\top \mathbf{x}_m(p)$, improvements can arise from (i) lowering unit prices $\mathbf{w}$, (ii) reducing resource use $\mathbf{x}_m(p)$, or (iii) increasing success probability $R_m(p)$. In practice, *cheaper tokens* reduce $\mathbf{w}$ via pricing changes, distillation, quantization, or optimized serving; *fewer tokens* reduce $\mathbf{x}_m(p)$ via prompt compaction, dynamic budgets, or instructions that promote concise generation; and *higher accuracy* increases $R_m(p)$ through better prompting, light test-time techniques, or improved model / training. Thus, the metric and framework capture these practical dimensions and quantify them in an interpretable way.

While these dimensions explain directional changes, the formulation of $v(m, p)$ still reports estimates (for cost and performance) solely through expectations and therefore does not capture variance. Two strategies with identical expected cost-of-pass may entail very different variances, and hence different risks. Similarly, extending the frontier cost-of-pass and gain (Equations (6) and (7)) to distributions (Equations (8) and (9)) also operates purely in expectation. Augmenting the metric with measures of variance or risk-adjusted objectives, or replacing expectation with alternatives such as median or worst-case values, could enhance interpretability and practical usefulness. We leave these extensions to future work.

## D.3 LIMITATIONS

We present limitations associated with both the framework and our evaluations. While covered in Section D.1, our evaluations instantiate simple formulations for costs and success. This is a reasonable proxy from a user perspective and extends gracefully, but it still omits indirect and context-specific terms (like evaluation/verification overheads, wait times, invocation retries, tool-call charges etc.). Our framework remains compatible with these terms via the vectorized cost view, but we do not include them in our core results.

Regarding the success metric, our framework assumes a binary success/failure criterion, thus continuous or composite notions of success are not modeled directly. This binary success modeling centers our measurements on the cost per unit of successful production. We believe this perspective

is especially useful in real-world domains where each successful unit of production has a reasonably similar relationship to the underlying economic objective. For example, in a support-center setting, a successful unit of production could be defined as whether a customer's query is resolved, with each resolved query contributing in a broadly similar way to the overall business goal. In scenarios that require more sensitive distinctions between different outcomes (for instance, when responses vary in quality or are evaluated along multiple dimensions), it may be beneficial to extend our framework to model cost per marginal unit of utility. The main idea would be to define a utility function over outcomes (potentially aggregating several quality dimensions into a single score) and study how much per unit of utility costs for a typical attempt. This utility could be expressed in dollar terms or in another unit, but the key idea is to make it continuous rather than purely binary, which can better capture complex objectives. Many high-stakes and complex domains, such as healthcare and finance place asymmetric value on different error types (e.g. false positives vs. false negatives), and such preferences can be encoded directly into the utility function. We leave a rigorous study of such formulations and their applications to future work.

Both pricing and performance can vary across API providers (Gao et al., 2025), especially for open-source models hosted by third parties. Treating each provider–model (i.e., inference pipeline) pair as a distinct strategy and either (i) reporting all results from the same provider consistently or (ii) providing multiple provider snapshots per model can make benchmarking and comparisons more robust.

Throughout our evaluations, we fix a single concise instruction and sampling arguments (e.g., temperature, top-$p$). We chose this to reduce degrees of freedom and enable comparability across models. However, results may be sensitive to these choices. Future work can study prompt and decoding sensitivity by evaluating small prompt ensembles per model and conducting sweeps over decoding settings.

Our empirical analysis focuses on widely used public benchmarks that serve as the community's current measures of progress. However, the evaluated models may have been partially exposed to these datasets (e.g. during pretraining). Therefore, these benchmarks are not a clean measure of out-of-distribution generalization. As a result, our findings are best interpreted as quantifying cost-effectiveness on canonical and widely familiar problems, and they may provide an optimistic view of the attainable economic frontier. In high-stakes deployments, or in any setting where data contamination is a major concern, practitioners should explicitly account for this potential training–test overlap.

Model selection in our evaluations can introduce temporal and categorical bias. Due to budget, compactness, and coverage considerations; we evaluated a subset of releases. For this, we fixed a short time window and chose representative models per major family to capture broad trends. A more exhaustive design is beyond our scope, but two extensions are natural: (i) broadening coverage to include historical and subsequent releases, and (ii) sampling more densely within a fixed horizon (more models at closely spaced release dates).

Our family distinction between lightweight and large models is based on per-token prices. Alternative categorizations (parameter count, open/closed status, deployment modality) are possible. We prioritize transparency and reproducibility; as sizes are often undisclosed, and openness does not map directly to user-incurred costs. We also keep the analysis prototypical by focusing on user-facing, common-case models (omitting quantizations or distillations). Future work can adopt alternative categorizations to quantify economic impact under different groupings.

The human expert baseline assumes that qualified annotators always succeed given sufficient time and compensation. In our experiments, we instantiate this by matching the expert with the benchmark's label creators and implicitly treating them as near-perfect. Extremely challenging problems (or domains with scarce expertise) may violate this assumption, and more realistic estimates of the human cost-of-pass would require dedicated human-subject studies that jointly measure success rates and labor costs. Our evaluations are further restricted to two strategy types: standalone model pipelines and the human expert baseline, omitting hybrid human-AI workflows in which models and humans jointly solve tasks and can outperform either component alone (Haupt & Brynjolfsson, 2025). Within our framework, such workflows can be represented as additional strategies (e.g. $m_{\text{hybrid}} \in \mathcal{M}$), with the frontier cost-of-pass computed over this expanded set. Accurately estimating

the cost-of-pass for these strategies would again require controlled human-subject studies that capture both AI inference and human labor, which we leave to future work.

Finally, our empirical conclusions about inference-time techniques are drawn under a canonical and common setup that is risk-neutral, where we primarily optimize dollar cost per unit of success. In domains with extreme penalties for errors, or where success is defined more stringently (for example, emphasizing reliability), our framework could instead favor strategies that employ heavier inference-time techniques to achieve very high reliability. Moreover, many such techniques can have non-trivial practical benefits (such as parallelizable majority voting (see Appendix D.1)) once richer cost and success models are adopted. A more extensive, risk-aware empirical analysis of these techniques is an important direction for future work.

Despite these caveats, the framework's abstract, modular design means each of the above extensions can be implemented by plugging in refined cost functions, richer success metrics, or additional variability terms. At the same time, our core analysis remains a practical baseline, as per-token API pricing reflects actual user-side costs, and binary pass/fail captures minimal utility in many applications. We hope future work adapts the framework along these lines and develops datasets that jointly stress cost and performance dimensions.

