# OpenReview forum: "Cost-of-Pass: An Economic Framework for Evaluating Language Models"
_ICLR.cc/2026/Conference — ICLR 2026 Poster_

### Official Review · Reviewer_bX6f · 2025-10-30

**Soundness:** 2
**Presentation:** 3
**Contribution:** 2
**Rating:** 2
**Confidence:** 4

**Summary:**

The paper proposes Cost-of-Pass, a metric for evaluating the economic efficiency of language models (LMs). The metric is defined as the expected monetary cost of producing a correct solution. The authors also define a frontier cost-of-pass, representing the minimum cost across available LMs and a human-expert baseline, and track how this frontier evolves over time and across model families (lightweight, large, reasoning). They report that lightweight models dominate simple tasks, large models excel at knowledge-based problems, and reasoning models are most efficient for complex quantitative reasoning. The work claims to ground LM evaluation in production theory and provide a principled economic framework for assessing progress.

**Strengths:**

The paper tries to address an increasingly relevant topic: economic efficiency and cost-awareness in LLM evaluation. The framework provides an intuitive and interpretable scalar metric that combines cost and accuracy. The evaluation also Includes multiple model families and task types, offering a broad empirical view. The use of “human-expert cost” as a reference baseline is a reasonable and communicable idea.

**Weaknesses:**

1. Theoretical novelty: The paper's formalism rephrases classic production frontier concepts without introducing new theoretical insights. The "cost-of-pass" is simply a ratio of cost per correct answer and the "frontier" is just the minimum of these values over models. This is an intuitive but elementary restatement, not providing any novel theoretical insights into the LLM industry. As a result, the so-called “economic foundation” adds rhetorical flavor but little intellectual substance.
2. Empirical metrics: The paper defines the cost purely as API pricing multiplied by token counts. However, these prices are commercial, provider-dependent, and change frequently by many economic factors. Thus, the analysis reflects vendor pricing strategies, not intrinsic model efficiency. The “economic” quantities are therefore non-fundamental and insightful. Also, “correctness” is defined as a binary pass/fail outcome per benchmark. This oversimplified notion of correctness ignores nuanced forms of reasoning quality, partial credit, or structured evaluation that are essential for emerging multimodal, interactive, or agentic tasks.
3. Experimental results: The empirical results primarily plot “frontier cost” over time or remove model families to show counterfactual trends. These are descriptive and unsurprising that lightweight models are cheaper for simple math, etc. The human-expert baseline is estimated from tutoring websites and contest durations, which introduces large uncontrolled variance. Most importantly, these insights have been largely discussed in previous work in cost-aware inference and routers and so on. The paper's results don't provide significant new insights.

**Questions:**

1. How can you model different tasks that LLMs are already really good at and are already solving in the real world?
2. How can your framework provide either machine learning insights or economics insights about future LLMs?

---

> ### Author Response · Authors · 2025-11-19
> **Official Response by Authors (1/2)**
>
> Thank you for the thoughtful review. Below are our responses to your comments:
>
> ---
>
> ### **W1:**
>
> We appreciate your thoughtful comments on the theoretical positioning of the paper. We agree that our goal is not to introduce a new economic theory, but to adapt well-established production theory into a grounded and operational evaluation framework for LMs. In response, we have revised the manuscript to clarify this scope and to soften language that may have overstated the theoretical novelty:
> - **Abstract:** We now explicitly state that our framework is “building on production theory” and change the wording from “introduce” to “formalize” to emphasize that we adapt an existing theory rather than propose a new one (lines 13-15).
> - **Introduction:** We now explicitly mention that our framework adapts ideas from production theory to the LLM setting (lines 96–97).
> - **Related Work:** We removed stronger claims about the impact and positioning of our framework (lines 456–457) and explicitly state that we adapt existing economic production principles (line 484).
>
> Conceptually, our adaptation treats a correct solution to a problem as the relevant “unit of output” and each LM (or inference pipeline) as a stochastic producer. In this setting, “cost-of-pass” corresponds to the “expected cost of producing one successful unit”, rather than an ad-hoc ratio. This expectation makes randomness explicit and directly answers the deployment question: “how many dollars do we expect to spend to obtain a correct solution with this inference pipeline?” Consequently, “taking the minimum over models” arises directly from the adaptation of production frontiers.
>
> This grounded perspective clarifies how prior cost-aware evaluations with ad-hoc heuristics differ from our framework. For example:
> - **(Wang et al., 2024)** evaluate reasoning strategies at a set of fixed compute budgets, reporting accuracy for each method at the same token or query limits. This describes how accuracy changes with budget under a fixed constraint, but does not directly address the deployment question of which budget-strategy combination is most cost-efficient.
> - **(McDonald et al., 2024)** propose the Economical Prompting Index (EPI), which multiplies accuracy by an exponential penalty on token usage controlled by a user-specified cost parameter C. The functional form and choice of C are specified heuristically rather than derived from an underlying economic model. This makes EPI scores harder to interpret in concrete economic terms or compare consistently across tasks, deployments, or against human workflows.
> - **(Nayab et al., 2024)** introduce conciseness-aware accuracy metrics that trade off explanation length against accuracy and latency. These metrics are defined in terms of output length and task-specific penalties, but do not extend directly to more generic deployment decisions that may also depend on factors such as pricing or tool usage.
>
> In contrast, our framework does not require fixing budgets, is grounded in an established theory, and is flexible enough to incorporate many real-world concerns, so that a single lens can organize a wide range of empirical findings. For example, using the same cost-of-pass formalism, we:
> - **Sec. 3.3:** track the velocity of progress over time and quantify that the frontier cost-of-pass on AIME-2024 has roughly halved every 7.1 months;
> - **Sec. 3.4 \& Sec. 3.5:** show that the economic progress in the LM ecosystem has been majorly made by the model-level innovations rather than the inference time techniques;
> - **Sec. C.2:** show on Tau-Bench that no single model family is currently essential for maintaining the frontier, suggesting that future gains on such tasks may require new forms of innovation.
>
> Plus, Sec. D.1 discusses how to extend the basic framework to more realistic scenarios by enriching the cost and performance modeling (lines 1190–1207), and illustrates this with an example of extending Tau-Bench into a real-world customer support setting (lines 1208–1224). These extensions are intended as evidence that our production-theoretic formalization can support richer and practically relevant analyses than existing ad-hoc metrics.
>
> **References:**
>
> 1. *Junlin Wang, Siddhartha Jain, Dejiao Zhang, Baishakhi Ray, Varun Kumar, and Ben Athiwaratkun. Reasoning in token economies: Budget-aware evaluation of llm reasoning strategies. arXiv preprint arXiv:2406.06461, 2024*
> 2. *Tyler McDonald, Anthony Colosimo, Yifeng Li, and Ali Emami. Can we afford the perfect prompt? balancing cost and accuracy with the economical prompting index. arXiv preprint arXiv:2412.01690, 2024.*
> 3. *Sania Nayab, Giulio Rossolini, Marco Simoni, Andrea Saracino, Giorgio Buttazzo, Nicolamaria Manes, and Fabrizio Giacomelli. Concise thoughts: Impact of output length on llm reasoning and cost. arXiv preprint arXiv:2407.19825, 2024.*

---

> ### Author Response · Authors · 2025-11-19
> **Official Response by Authors (2/2)**
>
> ### **W2:**
>
> We thank you for these insightful comments. You correctly note that API prices are shaped by commercial factors beyond a model’s intrinsic efficiency. However, we deliberately chose this metric for its key practical advantages:
> - Directly reflects the real-world dollar cost that practitioners and users face, makes our analysis immediately applicable.
> - Publicly verifiable, ensures transparency and reproducibility unlike internal or case-specific estimates.
>
> To help isolate technological progress from commercial adjustments, we use a single vendor for open models (for consistency) and anchor the analysis to each model’s earliest stable release price (Table 4 \& Sec. B).
>
> For the binary pass/fail outcome, our revision acknowledges this limitation in Sec. D.3. and discuss how it could potentially be extended to a continuous utility that captures nuanced forms of outputs (lines 1257-1278).
>
> ---
>
> ### **W3:**
>
> Thank you for the thoughtful feedback, helping us clarify our framework’s contribution.
>
> Beyond unifying known trends, our framework reveals more nuanced/novel results:
> - Formally shows that model-level innovations (Sec 3.4) are more cost-effective than many popular inference-time techniques (Sec 3.5),
> - Tau-Bench experiment (Sec. C.2) shows that each model family contains a cost-effective model (Table 7), and no family is essential for maintaining the frontier (Table 9); suggesting that further gains on this task will likely require new innovations (such as more agentic models / systems).
>
> For the human baseline estimation, the reviewer is right to note the inherent variance, even if our methodology in Sec. A uses a systematic “hierarchy of evidence” to ground the estimates in verifiable data. To assess robustness, we conduct a sensitivity analysis that re-estimates human-expert baseline costs using hourly rates in India as an alternative region (Sec. C.6), while keeping the qualification requirements \& time-per-question assumptions from Table 3 fixed. Under this considerably big change in pricing (Table 3 vs. Table 10), we observe that our findings still hold (Table 1 vs. Table 11), supporting the robustness of our estimation.
>
> ---
>
> ### **Q1:**
>
> We interpret this question in two ways \& address each.
>
> **How does the framework handle tasks where accuracy is already near-perfect?**
>
> This scenario highlights the usefulness of our framework. When accuracy saturates (Table 5) and no longer differentiates models (e.g. Basic Quantitative tasks) cost-of-pass shifts the comparison to cost-efficiency from raw performance. In this setting, lightweight models emerge as clear winners (Table 1), achieving the same near-perfect accuracy at a fraction of the cost (Tables 5 \& 6). Namely, our framework directly answers the practical question: “What is the cheapest way to solve this commoditized task?”
>
> **How can the framework identify which real-world tasks are economically viable for LMs now?**
>
> Our framework also provides tools to decide when LMs are economically attractive for deployment. Two relevant analyses:
> - **Frontier to human-baseline ratio.** The ratio between the LM frontier cost-of-pass and the human-expert baseline, ($V_p(M_t) / V_p(M_0)$), as in Figure 2, offers a task-level indicator of economic progress. A practitioner can set a viability threshold; a low ratio indicates that LMs already hold a substantial economic advantage over human experts.
> - **Essentialness.** As detailed in Sec. C.4, a 0\% essentialness score means that for every problem in the distribution, at least one LM is more cost-effective than a human. This signals that the task is fully economically dominated by LMs and highly viable for LM-based automation under the given cost model.
>
> ---
>
> ### **Q2:**
>
> We appreciate your question and see our framework as informing both ML and economic aspects on future LMs.
>
> First, by tracking the frontier cost-of-pass over time for each task (Figure 2), we can identify where progress is plateauing. Such plateaus suggest that further gains may require new paradigms rather than refinements of existing approaches.
>
> Second, the comparative analysis in Sec. 3.2 and Table 1 shows how different model families economically suit different task categories. Within a business’ own cost and performance modeling, the same type of analysis can support concrete decisions such as “avoid overpaying for cutting-edge models on commoditized, high-volume tasks” or “pay for frontier models only where failure is very costly.”
>
> Finally, our study on Tau-Bench (Sec. C.2, Table 9) shows that no single model family is strongly essential for achieving the current frontier, suggesting that future progress on such real-world tasks may depend on a new paradigm (e.g. improved agentic systems). Also, these estimates can inform economists about where current LMs are already close to being cost-competitive substitutes for human labor in specific domains \& where substantial gaps remain.

---

> ### Author Response · Authors · 2025-11-25
> **We Would Love to Hear From You!**
>
> Dear reviewer, thank you again for your thoughtful review. If you have any further comments or questions on our rebuttal, we would be very happy to address them.

---

### Official Review · Reviewer_GCEw · 2025-11-01

**Soundness:** 4
**Presentation:** 4
**Contribution:** 4
**Rating:** 10
**Confidence:** 4

**Summary:**

This paper posits the cost of generating a correct solution to a problem should be a central metric for understanding AI progress and it's potential economic impact. This can differ substantially from measures like price per token because the number of tokens generated for a response can vary substantially between models. However, it also differs from price per query because the model requires a correct answer and allows for many samples given some probability that one is correct. This is similar to a pass@k measure. Using this model they estimate price-at-pass costs for several quantitative tasks and also gather data on the cost of a human performing those tasks. This measure shows different niches for differing model sizes and non-trivial tradeoffs between accuracy and price. Further some inference time methods are valuable under this measure, and some like majority voting are useless (since the pass@k type measure would simply be able to select the correct answer from the majority voting system). They track this value over time showing rapid exponential decrease in cost-of-pass, but the rate of cost decrease is very non-uniform across their task domains.

**Strengths:**

- The paper poses an interesting metric that seems useful in understanding model improvement in a practical way
- Historical trends are analyzed over standard models and benchmarks, and human baseline costs are estimated

**Weaknesses:**

- The human price estimates do not seem very rigorously studied or calculated
- cost-to-pass has a number of limitations which the authors do address fairly well in Appendix D. I would love to see more discussion of what assumptions are needed for this metric to be a good reflection of reality and what domains that might hold true in.

**Questions:**

- I would love to see more discussion and exploration on the potential costs of finding/verifying the correct answer. (I now see there is some of this in Appendix D, I'd rather have it addressed in the main paper and even more consideration given in the appendix). It is especially unclear to me that Tau-bench in C.2 is applicable here. Wrong computer execution or airline bookings can be disastrous if a wrong trace is produced.
- It is very strange to me that your prices for MATH500 and AIME are so different. I believe these are extremely similar types of problems and skills involved and I would expect these to be performed by the same people and thus the tutoring wage should be the same.
- I'm pretty sure your method makes any majority vote or similar method trivially useless. Thus I don't see why you have both 3 and 4 player voting and so much exposition on it.
- I believe there is work by Epoch AI and Neil Thompson that seem relevant to this paper and the authors may be interested in looking into.

-- Regarding price performance increases in LLMS

Ben Cottier, Ben Snodin, David Owen, and Tom Adamczewski. LLM inference prices have fallen rapidly but unequally across tasks, 2025. URL https://epoch.ai/data-insights/llm-inference-price-trends. Accessed: 2025-09-03.

The Price of Progress
Hans Gundlach, Jayson Lynch, Matthias Mertens, Neil Thompson
https://openreview.net/pdf?id=JEsU87WUUb

-- regarding inference time compute methods

Pablo Villalobos and David Atkinson. Trading Off Compute in Training and Inference, 2023. URL145
https://epoch.ai/blog/trading-off-compute-in-training-and-inference. Ac-146
cessed: 2025-09-02.


-- regarding economic modeling of tasks and how to think about when they might be economically competitive with people

Beyond AI Exposure:
Which Tasks are Cost-Effective to Automate with
Computer Vision?
Maja S. Svanberg, Wensu Li, Martin Fleming, Brian C. Goehring, Neil C. Thompson

NBER WORKING PAPER SERIES
EXPERTISE
David Autor, Neil Thompson

Economic impacts of AI-augmented R&D
Tamay Besiroglu, Nicholas Emery-Xu, Neil Thompson

---

> ### Author Response · Authors · 2025-11-19
> **Official Response by Authors (1/2)**
>
> Thank you for the thoughtful review. Below are our responses to your comments:
>
> ---
>
> ### **W1:**
> > The human price estimates do not seem very rigorously studied or calculated
>
> Thank you for this important question. While our methodology tries to ground the estimates in verifiable data through a systematic “hierarchy of evidence”, we acknowledge that there are some factors that make our estimates appear non-rigorous (such as high variance in cost estimate ranges). To assess robustness, we conduct a sensitivity analysis that re-estimates human-expert baseline costs using hourly rates in India as an alternative region (Sec. C.6). We keep the qualification requirements and time-per-question assumptions from Table 3 fixed, and re-estimate only the hourly rates. Under this considerably big change in pricing (Table 3 vs. Table 10), we observe that our findings still hold (Table 1 vs. Table 11), supporting the robustness of our estimation.
>
> ---
>
> ### **W2, Q1:**
> > cost-to-pass has a number of limitations which the authors do address fairly well in Appendix D. I would love to see more discussion of what assumptions are needed for this metric to be a good reflection of reality and what domains that might hold true in.
>
> > I would love to see more discussion and exploration on the potential costs of finding/verifying the correct answer. (...) It is especially unclear to me that Tau-bench in C.2 is applicable here. Wrong computer execution or airline bookings can be disastrous if a wrong trace is produced.
>
> Thank you for raising these points. In the revision, we improve our discussion in Sec D.3 (lines 1264-1269) to clarify that our binary success metric is intended to reflect reality of domains where each successful unit of production has a similar economic effect (e.g. whether a support ticket is resolved), and we now explicitly emphasize in the subsequent sentences that handling domains with heterogeneous outcomes would require moving to a continuous utility formulation, whose rigorous study we view as a natural direction for future work.
>
> Moreover, while the first two paragraphs of Sec D.1 describe how richer deployment-level costs (such as verification effort and tool-use fees) can be incorporated into the cost representation and how the success metric can be augmented to capture different objectives (reliability vs. leniency), in the revision we add a new paragraph that illustrates how these ideas can be realistically applied in a potential extension of our canonical Tau-Bench evaluations to a real-world customer support center (lines 1208–1224).
>
> ---
>
> ### **Q2:**
> > It is very strange to me that your prices for MATH500 and AIME are so different. I believe these are extremely similar types of problems and skills involved and I would expect these to be performed by the same people and thus the tutoring wage should be the same.
>
> Thank you for the detailed examination! The price difference between AIME and MATH-500 stems directly from the difference in the required expertise and tutoring market. While AIME is a pre-Olympiad contest where every problem is difficult and requires niche, competition-specific strategies, MATH-500 is a broader sample from the full MATH dataset, spanning a wide range of difficulties (Levels 1-5). An average problem in MATH500 is therefore less complex, and can be effectively solved by a highly qualified general math tutor, rather than a specialized AIME coach. We realized that our description of the MATH500 dataset in Sec A was partially inaccurate. We made necessary revisions (lines 803-809), and reflected the distinction more clearly.
>
> ---
>
> ### **Q3:**
>
> > I'm pretty sure your method makes any majority vote or similar method trivially useless. Thus I don't see why you have both 3 and 4 player voting and so much exposition on it.
>
> We appreciate this observation. Majority voting and similar inference techniques are widely used in practice, so we think that understanding their behavior in our framework is important. In our canonical and risk-neutral experimental setting in Sec 3.5, the effective “dollars per correct answer” objective shows that a technique like majority voting does not justify its resource consumption for the accuracy gains (compared to other budget-oriented techniques like TALE-EP).
>
> However, we do not intend this to say that majority voting is useless in general. To better clarify this, we mention in the revision how the inference time techniques may become particularly useful, especially under different objectives such as reliability or scenarios where mistakes induce huge penalties (Sec D.3, lines 1320-1327). Moreover, specifically regarding majority voting, we share an example in Sec D.1 (lines 1231-1235) that when the objective expenditure is set to an alternative unit (such as wall-clock time), the parallelizability of executing majority voting could become very effective as the expected correctness would improve while incurring roughly the latency of a single vote.

---

> ### Author Response · Authors · 2025-11-19
> **Official Response by Authors (2/2)**
>
> ### **Q4:**
> > I believe there is work by Epoch AI and Neil Thompson that seem relevant to this paper and the authors may be interested in looking into. ...
>
> We sincerely appreciate all the closely related works, and we think that they help strengthen the positioning and grounding of our paper. In the revised manuscript, we cited them in the following format:
> - Section: related work, “Economic perspectives and broader impacts” subsection
>     - Beyond AI Exposure: Which Tasks are Cost-Effective to Automate with Computer Vision?
>     - NBER WORKING PAPER SERIES EXPERTISE
>     - Economic impacts of AI-augmented R\&D
>     - Trading Off Compute in Training and Inference
>     - **Positioning:** These papers analyze the economic impact of AI at a higher level of abstraction, with different lenses. Our framework supports such analyses by providing a micro-level, per-task based analytical tool.
> - Section: related work, “LM resource consumption, efficiency optimization and benchmarking.” subsection:
>     - LLM inference prices have fallen rapidly but unequally across tasks
>     - The Price of Progress
>     - **Positioning:** These works already conduct analyses that connect benchmark performance to monetary cost and chart historical trends. We build on this economic perspective in a complementary way by introducing our metric that operates at the level of individual problems and inference pipelines. This micro-level metric can be used inside their style of analyses.

---

> ### Author Response · Authors · 2025-11-25
> **We Would Love to Hear From You!**
>
> Dear reviewer, thank you again for your thoughtful review. If you have any further comments or questions on our rebuttal, we would be very happy to address them.

---

### Official Review · Reviewer_YBga · 2025-11-01

**Soundness:** 4
**Presentation:** 3
**Contribution:** 4
**Rating:** 8
**Confidence:** 4

**Summary:**

The paper introduces the cost-of-pass framework for evaluating the usefulness of language models in an economic light. They show that progress has been driven by lightweight and reasoning models for most domains. They also show that the impact of inference time techniques is nonexistent or minimal on the frontier cost of pass with the exception of cost-aware techniques like TALE-EP.

**Strengths:**

The paper engaged much more thoroughly with the economics than other literature. Particularly, evaluating human performance as part of the frontier cost of passing. I liked the attribution of progress to different model classes. It's interesting to see that reasoning models have led to progress adjusted for cost. I also thought the examination of inference time techniques was particularly interesting. Overall, a novel perspective, and I’d like to see more work like this engaging with economic constraints.

**Weaknesses:**

Ideally, the frontier cost of pass would include joint human+AI ability to solve problems. For instance, maybe a person with an AI chatbot can beat both another human and an AI chatbot. Human-AI collaboration ability is quite hard to measure experimentally, but some discussion of the literature in this area would be useful, ie, https://digitaleconomy.stanford.edu/wp-content/uploads/2025/06/CentaurEvaluations.pdf

I would probably try to summarize the related work section in the introduction (this is mostly already done), so I’d probably remove this section. There is a lot of mathematics to justify quite intuitive results. I’d probably try to move some of the exact definitions to the appendix and try to incorporate the limitations and further work in the main body of the paper. I’d also try to make the limitation section stronger (see the questions below).

**Questions:**

A more multidimensional discussion of inference time improvements would be helpful. Many techniques, for instance, majority vote, allow parallel processing, which dramatically increases cost but reduces wall clock time. What are the upsides to these techniques, or should people not use them in practice?

---

> ### Author Response · Authors · 2025-11-19
> **Official Response by Authors**
>
> Thank you for the thoughtful review. Below are our responses to your comments:
>
> ---
>
> ### **W1:**
> > Ideally, the frontier cost of pass would include joint human+AI ability to solve problems. For instance, maybe a person with an AI chatbot can beat both another human and an AI chatbot...
>
> Thank you for this insightful suggestion. We agree that human-AI collaboration is a crucial aspect of real-world AI adoption, and the true economic frontier may well be defined by these hybrid systems. In our revision, we discuss such an extension in lines 1308-1319.
>
> ---
>
> ### **W2:**
> > I would probably try to summarize the related work section in the introduction (this is mostly already done), so I’d probably remove this section. There is a lot of mathematics to justify quite intuitive results...
>
> Thank you for the thoughtful suggestions on the paper's structure and flow. We agree that making the paper more accessible is important. Based on your feedback, we took the following actions in the revision:
> - As you suggested, we enhanced the Introduction to more clearly summarize and contrast our economics-grounded framework with prior work (lines 52-54)
> - We have strengthened the limitations section (Sec. D.3) by incorporating the valuable feedback from the reviewers. However, we could not move this section to the main body, due to the page limits.
>
> On Mathematical Formalism (Sec. 2), we appreciate the goal of streamlining the paper. However, we respectfully believe that the formal definitions are the foundational core of our contribution. They establish the rigor and economic principles upon which our entire analysis is built. We worry that moving even parts of this logical foundation to the appendix may risk undermining the clarity and validity of our framework.
>
> ---
>
> ### **Q1:**
> > A more multidimensional discussion of inference time improvements would be helpful. Many techniques, for instance, majority vote, allow parallel processing, which dramatically increases cost but reduces wall clock time...
>
> Thank you for this excellent point. You are absolutely right that our main analysis, which focuses on dollar cost, does not fully capture the crucial, real-world aspects where such techniques would become valuable. We revised the manuscript to reflect such aspects better. In Sec. D.1, lines 1231-1235, we reflect your observation that majority voting can indeed be effective when the objective would be an alternative expenditure (wall-clock time). Moreover, in Sec. D.3, lines 1320-1327, we further dive deeper into when the inference time techniques would be useful and practical (such as under risk-aware settings), and acknowledge that our evaluation is under a canonical and risk-neutral setting.

---

> ### Author Response · Authors · 2025-11-25
> **We Would Love to Hear From You!**
>
> Dear reviewer, thank you again for your thoughtful review. If you have any further comments or questions on our rebuttal, we would be very happy to address them.

---

### Official Review · Reviewer_ucAV · 2025-11-01

**Soundness:** 2
**Presentation:** 3
**Contribution:** 3
**Rating:** 4
**Confidence:** 4

**Summary:**

This paper proposes an economic framework for evaluating language models that jointly considers accuracy and inference cost. The authors introduce "cost-of-pass” as the expected monetary cost to generate a correct solution and "frontier cost-of-pass”. The framework is applied to analyse 11 language models across 6 benchmarks. Results indicate that: 1) different model families (lightweight, large, reasoning) are cost-effective for different task types, 2) frontier cost-of-pass has decreased exponentially, particularly for complex quantitative tasks and 3) model-level innovations drive progress more than inference-time techniques.

**Strengths:**

1. The paper has strong theoretical grounding in established production theory.
2. It is clearly formulated and easy to follow, even for those working in parallel fields.
3. It has practical and timely relevance as models become increasingly expensive and real-world decisions become more critical.
4. Experiments are comprehensive.

**Weaknesses:**

1. There seems to be no discussion or handling of (KV or similar) caches. These can significantly reduce the cost of inference. Similarly, no discussion of batching opts that have large cost effects. These are critical limitations that need to be highlighted.
2. The binary assumption does limit the practical application. There is no discussion or proposal on how this might apply when precision/recall trade-offs are important. Related to this, no discussion of how different domains might care about different error types, such as healthcare vs finance vs research. The authors partly acknowledge this but I think the paper would be much stronger if it was given more comprehensive consideration.
3. Similarly, the quality of the answer can have a significant difference. If two models both give the right answer but only one of them gives detailed explanations with in-depth knowledge, that might be much more valuable than the other simple answer. Another limitation that isn’t discussed sufficiently.
4. The cost-of-pass might be falling not because models are more efficient, but because the problems are getting stale and contaminated in training data. Models (including smaller/cheaper ones) can get “unnaturally” high scores on some benchmarks because they target them specifically. What is the risk that MATH-500 is increasingly used in training data for newer, smaller models, artificially increasing their score on them?
5. It is not clear to me if the framework handles amortisation in real LM deployment sufficiently. In classical production making 10 products = 10x cost of 1 product (in the same factory). But LM deployment is more dynamic when it comes to problem solving, each one is unique but can leverage learnings from previous ones to improve performance, e.g. through prompt improvements. Can these be assumed to be approximately the same? This thought/question is partly inspired by Learning by Doing by Kenneth Arrow.
6. Writing sometimes feels like patch-work. Abbreviations like “language models (LMs)” are defined three times. This only needs to be done once.
7. No analysis of how sensitive the conclusions are to variations in human-expert cost estimates. Given the wide variability in expert compensation ($15-$100/hour for different expertise levels, per Appendix A), this might be significant.
8. No confidence intervals or standard deviations reported for cost-of-pass estimates. Hard to judge variance and significance.

**Questions:**

1. Being a little disingenuous here: what would a naive baseline model that is one random selection call (~$0.00001 per attempt) score as a cost-of-pass? Does there need to be some weighting or other term added since the low cost term dominates as the cost tends to 0, especially in MCQ tasks?
2. The paper assumes Rexpert(p) ≈ 1 (line 190), but provides no empirical validation that human experts actually achieve near-perfect accuracy on these benchmarks. Where does this assumption come from?
3. If a model consistently fails on certain problem types, repeated attempts won't help, so won't 1/Rm(p) be an overestimate of required attempts? These aren’t independent tests.
4. How were the 6 benchmarks selected? Why not include code generation, long-context, or creative tasks?
5. Which specific versions of each model were used? (GPT-4o has had multiple versions)
6. Have the authors considered median or worst-case cost-of-pass instead of expectation? This might be useful in many fields.

---

> ### Author Response · Authors · 2025-11-19
> **Official Response by Authors (1/2)**
>
> Thank you for the thoughtful review. Below are our responses to your comments:
>
> ---
>
> ### **W1:**
> Thank you for this excellent and practical point. In our revision, we now highlight how to incorporate caching (as an amortized, one-time setup cost spread across inference) and batching (as an adjustment to input and output token costs under the chosen batching configuration) in Sec. D.1 (lines 1197-1200).
>
> ---
>
> ### **W2, W3:**
> Thank you for these insightful comments; we have added more discussions into the revision considering different error types and answer qualities. While our experiments use “correctness” as a binary performance metric, Sec. D.1 (lines 1201-1207) explains how alternative binary success criteria can be substituted into our framework, including ones that can embed precision/recall trade-offs (lines 1205-1207). We also more clearly acknowledge the binary assumption as a limitation, and in Sec. D.3 (lines 1263-1278) we suggest extensions to continuous or more fine-grained performance indicators. These extensions provide a way to encode different error costs across domains such as healthcare and finance (lines 1275-1277). Similarly, in lines 1268-1273 we discuss how richer notions of answer quality could be incorporated by redefining the notion of “success” under such suggested extensions.
>
> ---
>
> ### **W4:**
> Thank you for raising this critical point, as contamination on public benchmarks like MATH-500 is a significant concern with important implications for interpreting empirical results. We acknowledge this concern in Sec. D.3, lines 1289-1295.
>
> ---
>
> ### **W5:**
> Thank you for raising this perspective. We agree that LM deployments can evolve over time through prompt and pipeline changes rather than remaining fully static. Our empirical analyses therefore consider deployments as static snapshots: a given model together with its inference pipeline (e.g. prompt, decoding) evaluated on a distribution of tasks. We adopt this snapshot view because, in typical real deployments, prompt and pipeline updates occur on a slower, batched timescale than individual queries. Thus, between updates, the system serves many queries with approximately fixed success probability and expected per-attempt cost. For example, in Sec. 3.3 we recompute the frontier at each release, treating each model release as a separate snapshot.
>
> Within this framework, a more dynamic system can be represented by evaluating particular snapshots along its evolution, such as the state at time t after a sequence of prompt and pipeline updates. The cost of such a snapshot can incorporate both usage costs and amortized fixed costs. Concretely, a practitioner can aggregate the one-time investments required to reach that state (e.g. engineering effort, tuning runs, infrastructure costs) and divide them by the expected number of queries served in that state, adding this amortized amount to the per-token usage cost in $C_m(p)$ as discussed in Sec. D.1.
>
> Such dynamic and sequential treatment of evolution is beyond the scope of this paper, but we view our framework and its amortised-cost formulation as the basic economic lens for such analyses.
>
> ---
>
> ### **W6:**
> Thank you for the careful reading and for providing valuable feedback on the writing. We identified a repeated definition of “language models (LMs)” (line 150) and, in the revision, simplified it to “LMs.” Additionally, we performed a full review of the paper to improve the writing quality, and we found and corrected a typo in the formulation on line 1192.
>
> ---
>
> ### **W7:**
> Thank you for raising the point regarding sensitivity of human baseline estimation. While the human expert baseline has noticeable inherent variance, to assess robustness, we conduct a realistic sensitivity analysis that re-estimates human-expert baseline costs using hourly rates in India as an alternative region (Sec. C.6). We keep the qualification requirements and time-per-question assumptions from Table 3 fixed, and re-estimate only the hourly rates. Under this considerably big change in pricing (Table 3 vs. Table 10), we observe that our findings still hold (Table 1 vs. Table 11), supporting the robustness of our estimation.
>
> ---
>
> ### **W8:**
> Thank you for raising this point about the statistical significance of our results. In the revision, we report 95\% bootstrapped confidence intervals (10,000 samples) in Sec. C.7. As shown in Table 12, using the midpoints of these intervals as estimates largely reproduces the rankings in Table 1. The only notable change is that on GPQA Diamond, o1-mini attains a lower frontier cost-of-pass than Llama-3.1-405B. Overall, we believe this showcases the significance of our findings.

---

> ### Author Response · Authors · 2025-11-19
> **Official Response by Authors (2/2)**
>
> ### **Q1:**
> Thank you for this insightful thought experiment. We see this as an important point regarding applicability and reliability of our framework in real-world deployments. In the revision, we conducted a new experiment (Sec. C.8) where we considered a realistic scenario: treating a lightweight model (i.e. Llama-3.1-8B) as a random caller due to either its hypothetically naive or even malicious intent under our MCQ tasks (BBQ and GPQA-Diamond). Our results in Table 13 highlight that such naively acting lightweight baseline can achieve extremely low frontier cost-of-pass, claiming to have made significant economic progress over human-expert baseline while actually it is naive and not practically useful at all. The details of the experiment design and the motivating formulations are in lines 1121-1173. We also discuss and suggest how so-called “benchmark hacking” scenarios could be mitigated in lines 1174-1181.
>
> ---
>
> ### **Q2:**
> Thank you for the important question. We agree this is an approximation and a valuable point of discussion. We realized that in our manuscript, this assumption has not been clearly presented. In the revision, we added a footnote into Sec. 2.4 where we briefly share the justification behind our assumption, and refer the readers to Sec. D.3 (lines 1308-1313) where we elaborate further on the basis of this assumption and its limitations.
>
> ---
>
> ### **Q3:**
> This is an important point that highlights a key nuance of the framework. You are correct that if a model has a fundamental capability gap, simply repeating attempts will not lead to success. Our framework is designed to capture this exact scenario. The term $R_m(p)$ represents the model's fixed probability of success on a given problem. The "independence" we assume is in the sampling process for each attempt, not an independence from the model's underlying (and fixed) capabilities (encapsulated under m). In the exact case you describe (a model that consistently fails) the empirically measured success rate $R_m(p)$ will be 0. Consequently, our cost-of-pass metric ($C_m(p) / Rm(p)$) correctly becomes infinite. This is the desired outcome, as it formally signals that this model cannot solve the problem, no matter how much economic resources we pour into it Conversely, when a model is capable but stochastic ($0 < R_m(p) < 1$), the $1/R_m(p)$ term correctly models the expected number of attempts required to achieve the first success due to sampling randomness.
>
> ---
>
> ### **Q4:**
> Thank you for the question about the scope of our evaluation. Our primary goal was to demonstrate the framework's analytical power across a complementary spectrum of tasks that community reports progress in. Each category for the tasks was chosen to reveal distinct economic trade-offs that standard metrics (like accuracy) might miss. For example, in basic quantitative tasks, where accuracy is often saturated, our framework moves beyond a simple pass/fail to highlight the crucial role of cost-efficiency, demonstrating the economic advantage of lightweight models. Regarding the suggestions, while our analysis does not incorporate code generation or creative tasks, we included a complete analysis on Tau-Bench in Sec. C.2, which explicitly evaluates models on multi-turn (long-context), tool-use interactions (agentic), testing the framework in a practical setting.
>
> ---
>
> ### **Q5:**
> Our methodology was to evaluate the earliest stable version of each model (lines 299-301) corresponding to the release dates and pricing listed in Table 4. This strategy helps our temporal analysis (e.g. Figure 2) to accurately capture the economic impact of each major model release to the ecosystem.
>
> ---
>
> ### **Q6:**
> Thank you for this thoughtful question. In our framework, we use the expectation operator to find the costs associated with per sampled response from the model for a given problem. Moreover, we also use the expectation operator to aggregate the frontier cost-of-pass and gain formulations (Sec. 2.5, Eqs. 8-9) from individual problems to a benchmark-level statistic. We think the alternative aggregators are considerable, and in our revision, we mention and discuss them in lines 1246-1253.

---

> ### Author Response · Authors · 2025-11-25
> **We Would Love to Hear From You!**
>
> Dear reviewer, thank you again for your thoughtful review. If you have any further comments or questions on our rebuttal, we would be very happy to address them.

---

### Author Response · Authors · 2025-11-19
**Final Remarks and Summary of Revisions**

Dear Reviewers and ACs,

We appreciate your efforts and valuable feedback! As a recap of reviews and our revisions, we first share the strengths and merits of our work as identified by reviewers, then summarize the changes we made in response to the reviewer comments.

### Strengths of our work as identified by reviewers
- **Conceptual framing and metric:** Reviewers found our work’s novel perspective on engaging with economic constraints to be motivating for further research (*YBga*), and described the proposed metric as interesting (*GCEw*), intuitive and interpretable (*bX6f*). They also noted its usefulness for understanding model improvement in a practical way (*GCEw*).

- **Theoretical grounding and economic perspective:** Our framework was recognized as having strong theoretical grounding in established production theory (*ucAV*) and as engaging more thoroughly with economics than other literature in this space (*YBga*). The inclusion of human performance as part of the frontier cost-of-pass was positively received (*YBga, GCEw, bX6f*), with one reviewer noting it a reasonable and communicable idea (*bX6f*).

- **Clarity and practical relevance:** Reviewers highlighted the clear and easy-to-follow formulation (*ucAV*), and the work’s practical and timely relevance for increasingly expensive models and critical real-world cost-performance trade-offs (*ucAV, bX6f*).

- **Empirical coverage and insights:** Experiments were described as comprehensive (*ucAV*), with useful and interesting takeaways such as attributing progress to different model classes and examining inference-time techniques (*YBga*). Reviewers also appreciated that the study includes multiple model and task types, offering a broad empirical view (*bX6f*), and positively perceived the analysis of historical trends over standard models and benchmarks (*GCEw*).

### Summary of our revisions in response to reviewer comments
In addition to addressing the reviewers’ questions and weaknesses, we made the following revisions to further strengthen the manuscript (highlighted in blue):

- **Toning down claims of theoretical novelty:** We clarified that our main contribution is adapting production theory into a grounded, operational evaluation framework for LMs, and softened language (in Abstract, Introduction, and Related Work) that could be read as overstating theoretical novelty. (*bX6f*)

- **Writing and presentation improvements:** We strengthened the introduction by bringing key related work earlier, expanded Related Work with additional recommended papers, and corrected minor issues such as typos and redundant definitions (e.g. repeated definition of “language models”). (*ucAV, GCEw*)

- **Human expert baseline:** We clarified the basis of our human baseline assumptions in the main text (Section 2.4) and Section D.3, corrected the cost estimation for MATH500 and clarified its distinction from AIME in Section A, and added a regional ablation (using hourly rates from India) as a sensitivity analysis of the human expert baseline (*bX6f, ucAV, GCEw*)

- **Confidence intervals:** In Section C.7, we now report 95\% bootstrapped confidence intervals for Table 1. (*ucAV*)

- **Benchmark hacking scenario:** In Section C.8, we added an adversarial/naive baseline experiment to illustrate how “benchmark hacking” scenarios can arise, affect interpretation, and be mitigated. (*ucAV*)

- **Extensions in Section D.1 for complex deployment modeling:** We enriched Section D.1 to show how the cost function can model caching and batching as amortized operations, how alternative binary success criteria can encode precision/recall trade-offs, how our framework can instantiate a realistic Tau-Bench-extending customer-support setting, and how the majority voting’s parallelizability become appealing when latency, rather than dollar cost, is the primary objective. (*ucAV, YBga, GCEw*)

- **Alternatives to the expectation operator:** In Section D.2, we note that alternative aggregators (such as median or worst-case) may be appropriate when modeling distributions of cost or cost-of-pass under different real-world objectives. (*ucAV*)

- **Limitations and future directions in Section D.3:** We expanded the limitations discussion to clarify the scenarios our framework is most appropriate for, describe how future work could extend binary success modeling to continuous utilities, and show how different error types and domain-specific costs can be encoded. We acknowledge contamination concerns on public benchmarks, elaborate on the human-baseline's near-perfect accuracy assumption and its limitations, highlight hybrid human-AI systems as a promising direction, and note that while our experiments use a canonical risk-neutral setting, inference-time techniques are valuable for risk-aware or reliability-focused deployments. (*bX6f, ucAV, YBga, GCEw*)

We hope these revisions address the reviewers’ concerns, clarify our scope and contributions, and strengthen the manuscript.

---

### Meta-Review · Area_Chair_QpF9 · 2026-01-07

**Summary:**

The paper introduces the idea of ‘cost-of-pass’ as a novel LM eval approach which is the expected monetary cost for a model to produce the correct answer. The framing around the cost to generate a correct solution ties into understanding progress and economic value. The empirical insights into problems, model families and test-time techniques is timely given the interest in the economic value of LLMs

There was divergence among the reviewers: 2 strong accepts and 2 rejects. The main disagreement wasn’t around the metric being useful or timely or around experiments, rather around the theoretical novelty of the production theory concept and using api costs as the proxy.

Recommendation: Given the lack of reviewer engagement and on the basis of the authors having provided a thorough rebuttal, additional experiments and paper updates to address the reviewers concerns (both with new experiments and positioning changes based on the reviewer comments), I believe the paper will bring value to the ICLR community and spur discussion. Hence, vote to accept the paper

**Reviewer Concerns:**

see "Summary" text

**Reviewer Scores:**

see "Summary" text

---

### Decision · Program_Chairs · 2026-01-26

Accept (Poster)